# Modulatory dynamics of periodic and aperiodic activity in respiration-brain coupling

Daniel S. Kluger [1,2] ✉, Carina Forster [3,4,5], Omid Abbasi[1], Nikos Chalas[1,2], Arno Villringer [3,4,6] & Joachim Gross [1,2]

Bodily rhythms such as respiration are increasingly acknowledged to modulate neural oscillations underlying human action, perception, and cognition. Conversely, the link between respiration and aperiodic brain activity – a non-oscillatory reflection of excitation-inhibition (E:I) balance – has remained unstudied. Aiming to disentangle potential respiration-related dynamics of periodic and aperiodic activity, we applied recently developed algorithms of time-resolved parameter estimation to resting-state MEG and EEG data from two labs (total $N = 78$ participants). We provide evidence that fluctuations of aperiodic brain activity (1/f slope) are phase-locked to the respiratory cycle, which suggests that spontaneous state shifts of excitation-inhibition balance are at least partly influenced by peripheral bodily signals. Moreover, differential temporal dynamics in their coupling to non-oscillatory and oscillatory activity raise the possibility of a functional distinction in the way each component is related to respiration. Our findings highlight the role of respiration as a physiological influence on brain signalling.

Recent years have seen a surge in reports of respiration as a modulator of brain signalling - at least partly through modulation of rhythmic brain activity[1]. These respiration-modulated brain oscillations (RMBOs) have been localised within a wide-spread network across the cortex[2] and are increasingly being linked to changes in motor function[3,4], cognitive processing[5–7], and fundamental aspects of perception[8,9]. One candidate mechanism by which respiration (and other bodily signals such as the cardiac rhythm; see e.g.[10]) is thought to coordinate brain signalling is cortical excitability, a dynamic brain state indexing neural populations' readiness to become activated. This proposal rests on findings from intracranial animal studies demonstrating that respiration modulates spike rates in a variety of brain regions[11–13]. Moreover, one recent study[9] directly investigated respiratory modulation of excitability during low-level visual perception. Critically, brain activity is not entirely oscillatory:

One can distinguish its genuinely periodic, oscillatory component from an underlying stationary, aperiodic component typically showing a characteristic 1/f dependence of power and frequency (meaning that lower frequencies carry higher power[14]). In fact, dominance of low-frequency power is by no means just an epi-phenomenon of neural power spectra, but does reflect meaningful characteristics of brain function. Most relevantly, the 'steepness' of the aperiodic component has been shown to reflect the dynamics of excitation-inhibition (E:I) balance[15–17], with steeper 1/f slope indicating a stronger influence of inhibitory activity. While sophisticated algorithms exist to disentangle periodic and aperiodic components of neural power spectra[18], time-resolved estimations of 1/f slope in particular have proven computationally expensive and lacked empirical validation. Consequently, the selective dynamics linking peripheral signals (such as respiration) to

[1]Institute for Biomagnetism and Biosignal Analysis, University of Münster, Münster, Germany. [2]Otto Creutzfeldt Center for Cognitive and Behavioral Neuroscience, University of Münster, Münster, Germany. [3]Department of Neurology, Max Planck Institute for Human Cognitive and Brain Sciences, Leipzig, Germany. [4]Charité – Universitätsmedizin Berlin, Einstein Center for Neurosciences, Berlin, Germany. [5]Charité – Universitätsmedizin Berlin, Bernstein Center for Computational Neuroscience, Berlin, Germany. [6]Humboldt-Universität zu Berlin, Faculty of Philosophy, Berlin School of Mind and Brain, Mind-BrainBody Institute, Berlin, Germany. ✉e-mail: daniel.kluger@uni-muenster.de

periodic and aperiodic neural activity are entirely unstudied. Meanwhile, the recent introduction of the SPRiNT toolbox[19] provides methodological advances that allow us to parameterise stationary and oscillatory brain activity dynamically over time. Here, we apply these algorithms in the realm of human body-brain interactions to test the hypothesis that 1/f slope as a marker of E:I balance is modulated by respiration.

Specifically, we used concurrent resting-state M/EEG and respiratory recordings in populations from two labs (Münster, Leipzig) to investigate distinct respiratory modulations of periodic and aperiodic brain activity. Our rationale was twofold: First, we aimed to characterise respiration phase-locked changes of aperiodic 1/f slope over parieto-occipital cortices to corroborate previous findings of excitability changes in visual perception[9]. Second, we extended the scope for a general investigation of potentially independent modulations of aperiodic and oscillatory components within the cortical RMBO network[2]. Here, we particularly aimed to assess whether both components of neural signalling would be modulated by respiration and, if so, how these modulatory dynamics would be related to one another. Our results highlight respiration as a slow physiological modulator of both periodic and aperiodic cortical signalling, strengthening its previous implication in the regulation of excitability states. Furthermore, the over all pattern of findings strongly suggests differential temporal dynamics between the coupling of both components.

## Results

We computed time-resolved fits of both aperiodic (1/f) and periodic (oscillatory) components of resting-state brain activity to disentangle their dynamics in respiration-brain coupling. A moving-window application of the novel SPRiNT algorithm[19] and the underlying specparam procedure[18] to MEG and EEG data yielded individual time

courses of aperiodic slopes and periodic spectra for a total of $N = 78$ participants. Concurrent respiration recordings allowed us to extract respiratory phase vectors that corresponded to each of the moving windows. Next, we performed binning of respiratory phase and bin-wise averaging of corresponding 1/f slope and periodic spectra on the group level (see Fig. 1 and Methods for details). We then used these phase-specific estimates of slope and periodic spectra to statistically test if and how periodic and aperiodic brain activity is modulated by respiration.

### Respiration relates to aperiodic neural activity over posterior cortices

The first analysis was closely connected to a recent study showing respiration phase-dependent modulations of cortical excitability in a visual detection task[9]. In line with a rich literature of visual perception studies (see[20]), increased excitability was indexed by desynchronised alpha oscillations (i.e., lower alpha power corresponding to higher excitability). However, changes in alpha power can be confounded by changes in the slope of aperiodic background activity[18] - itself a widely-used measure of cortical excitability and excitation/inhibition balance[17]. Therefore, we aimed to disambiguate the effects of respiration on periodic and aperiodic brain activity and started by first testing the hypothesis that respiration-induced changes of excitability are reflected in cyclic changes of the slope of aperiodic brain activity. For both MEG (Münster) and EEG data (Leipzig), we set up a linear mixed-effects model (LMEM) to predict individual 1/f slopes of parieto-occipital average power spectra by respiration phase. A null distribution of the resulting beta weights was constructed by re-computing the LMEM based on individual surrogate respiration time series $k = 5000$ times. Subsequently, we pooled both data sets in order to capitalise on

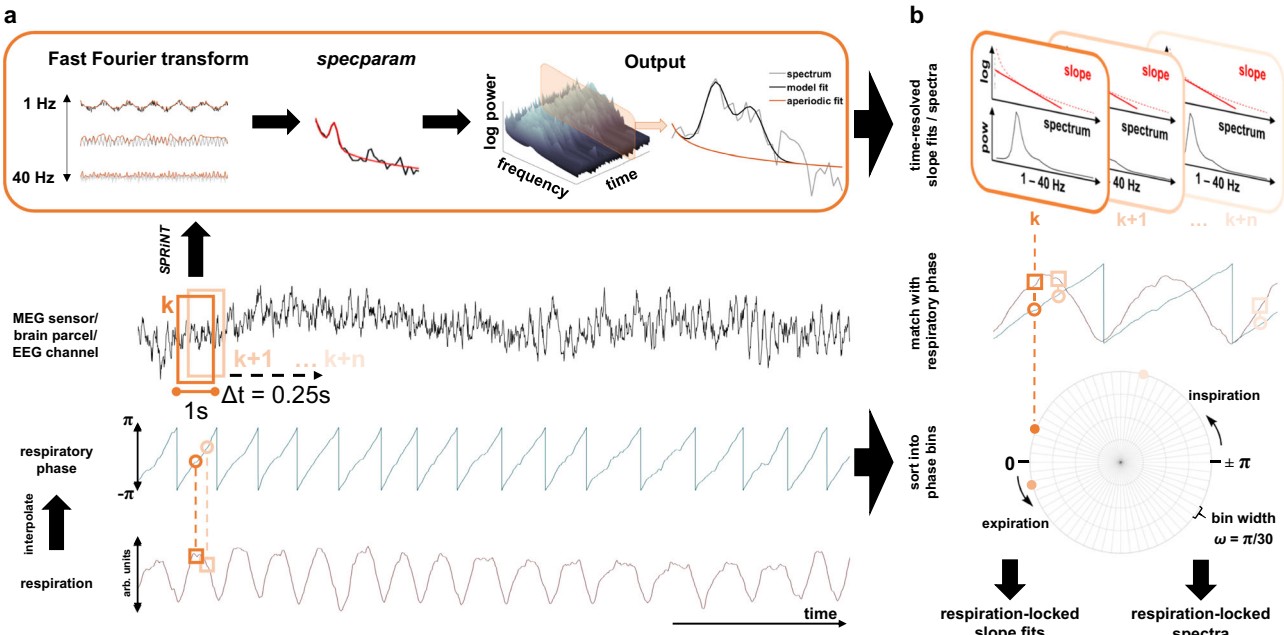

**Fig. 1 | Synopsis of acquired data and applied methods. a** In two labs, we simultaneously acquired nasal respiration as well as eyes-open, resting-state MEG (IBB, Münster) and EEG data (MPI, Leipzig) in continuous 5-min recordings. After preprocessing, single-sensor/-channel M/EEG data (middle panel) were subjected to the SPRiNT algorithm[26] (top). Here, using a moving-window approach (window length = 1 s, 75% overlap between neighbouring windows), spectral components of neural time series are estimated using a Fast Fourier transform. These frequency-domain data are then parameterised using the specparam algorithm[18] which yields both aperiodic and periodic components of neural activity in that time window. Repeating this procedure along the entire recording thus yields time-resolved fits of the aperiodic 1/f slope as well as time-resolved periodic spectra ranging from

1–40 Hz (top right). Respiratory phase was computed using two-way interpolation (int) of the normalised raw respiration signal (peak-to-trough, trough-to-peak; bottom). **b** For each time point used as a moving-window centre in the SPRiNT algorithm, we then extracted the corresponding respiratory phase. This allowed us to sort all time-resolved slope fits and periodic spectra according to the respiratory phase at which they had been computed. In keeping with previous work[2,9] we finally partitioned the respiration cycle into $n = 60$ equidistant, overlapping phase bins and computed bin-wise averages of slope fits and periodic spectra. This approach thus yielded quasi-continuous, respiration phase-resolved courses of both periodic and aperiodic components of brain activity for each sensor/channel within each participant (bottom right).

higher statistical power and to illuminate phase-locked modulations jointly found across modalities. For both MEG and EEG as well as for the pooled data, the empirical group-level beta weight for respiratory phase was significantly higher than the respective null beta weights (all $p < .001$), strongly indicating a significant over all association between respiratory phase and aperiodic neural signals over posterior cortices (Fig. 2). In order to characterise the temporal dynamics of respiration phase-locked changes in 1/f slope, we repeated the binwise slope estimation as outlined above, but this time on surrogate respiration time courses. For each participant, we thus generated $k = 5000$ 'null slope estimations' within each phase bin which we used to extract percentiles for the empirical group-level bin averages relative to these binwise 'null averages' (Fig. 2a; see Methods for details). For the pooled data from both labs, we observed a steeper slope (indicating stronger inhibitory activity) around the expiration-to-inspiration transition (ranging from -156° to 143° and around 131°). Conversely, the slope was flatter (indicating stronger excitatory currents) during the inspiratory phase (between -88° and -64°) and shortly after the inspiration-to-expiration transition (9° to 46°, see Fig. 2). This general pattern was consistent across both data sets, although the flattening of the slope during the inspiratory phase did not reach significance in the EEG data alone (presumably due to slightly lower statistical power). Consistency across MEG and EEG data was supported by a non-parametric two-sample test for circular data (akin to a Kruskal-Wallis test for linear data), which showed that the circular means of MEG and EEG slope-respiration courses were not significantly different from one another ($P(77) = 0.21$, $p = .651$).

## Aperiodic modulations as a function of respiratory parameters
We conducted different control analyses to assess whether 1/f slope and its modulation strength covaried with parameters like breathing rate, depth, or route (i.e., nasal vs oral breathing). First, in the present data, individual 1/f slopes (averaged across the respiratory cycle) were not significantly correlated with breathing rates (MEG: $r(40) = -.17$, $p = .297$; EEG: $r(38) = .13$, $p = .418$) or breathing depths (MEG: $r(40) = .19$, $p = .230$; EEG: $r(38) = -.09$, $p = .591$) in either data set (Fig. 2c). However, the interpretability of these results is somewhat limited by the small amount of intraindividual variation in the respiratory time series - natural breathing is simply too consistent to conclusively investigate whether different breathing 'modes' (e.g., deep vs normal breathing) would influence respiration-locked modulations in 1/f slope. Therefore, we next re-analysed a previously published data set[3] in which we had recorded 5 min of whole-head resting-state MEG during both deep and normal nasal breathing (see Fig. 2d). In these data, a comparison of deep vs normal nasal breathing revealed that 1/f slope over the posterior ROI was indeed more strongly modulated during deep (compared to normal) breathing ($z(27) = 3.02$, $p = .003$, see Fig. 2e). Finally, to investigate the role of breathing route, we compared respiration-locked 1/f slope modulations at rest during nasal vs oral breathing in a new sample of $N = 25$ MEG data sets (see Methods). Repeating the posterior ROI analysis described above, respective LMEMs confirmed significant coupling of respiratory phase to 1/f slope during both nasal and oral breathing (both $p < .001$; Fig. 2f). The modulatory pattern over phase was again similar to the original findings (with certain limitations due to reduced power): During nasal breathing, 1/f slope was flattened during the inspiratory phase (-95° to -52°) as well as shortly after the inspiration-to-expiration transition (around 36°) and steeper around the expiration-to-inspiration transition (143° to -150°). During oral breathing, 1/f slope was flattened around the inspiration-to-expiration transition (-64° to 52°) and steeper around the expiration-to-inspiration transition (131° to -125°). A comparison of individual ranges of slope - respiration (i.e., $max_{slope} - min_{slope}$ over the respiratory cycle) revealed that the strength of respiration-locked 1/f slope modulations did not significantly differ between nasal and oral breathing ($z(24) = 0.80$, $p = .426$).

## Respiration-locked slope modulations are wide-spread, but strongest over posterior sites
Overall, the first analysis demonstrated a consistent pattern of respiration-locked 1/f slope modulations over parieto-occipital sensors: The slope of aperiodic neural activity was not uniform over the respiratory cycle, but systematically modulated in such a way that 1/f slope was flattened around the inspiration-to-expiration transition and steeper around the expiration-to-inspiration transition. In the next step, we aimed to (i) broaden the scope to investigate respiration-locked slope changes across all M/EEG sensors and (ii) characterise both direction and strength of these modulations across the scalp. For each participant, single-sensor/channel 1/f slope courses across 60 respiratory phase bins were extracted using the pipeline described above on single-sensor time series. For each of the $N = 275$ MEG sensors (Münster) and $N = 62$ EEG channels (Leipzig), we then repeated the combined LMEM and permutation approach (see above). Thus, for each sensor, the percentile of the empirical beta weight (relative to its null distribution constructed from LMEMs on surrogate respiration) indicated significance of phase-locked 1/f slope changes. Group-level average slope courses for $n = 240$ significant MEG sensors (top) and $n = 53$ significant EEG channels (bottom) are shown in Fig. 3a. To quantify whether there was a consistent modulation of 1/f slope across all sensors/channels, we computed the circular mean of single-sensor/channel slope courses. The histograms in Fig. 3a illustrate a clear clustering around the expiration-to-inspiration transition (i.e., phase bins around $\pm \pi$). Supporting our parieto-occipital findings from above, Rayleigh tests on sensor-/channel-wise circular means corroborated that single-sensor/channel slope courses were not uniform across respiratory phases, but instead showed a consistent mean direction with the steepest slope around $\pm \pi$ for both data sets (MEG: $z = 126.86$, $p < .001$; EEG: $z = 14.38$, $p < .001$). A topographic representation of sensor-/channel-wise circular means is shown in Fig. 3b.

Finally, in line with previous approaches[2,7], we quantified the strength of single-sensor/channel phase-locked slope modulations relative to their null distributions: For each sensor/channel, we computed z-scored LMEM beta weights by subtracting the mean of its 'null beta' distribution and dividing by the corresponding standard deviation (see Methods). In both MEG and EEG topographies, the strongest modulations of aperiodic neural activity were indeed observed over parieto-occipital cortices (Fig. 3c).

## Periodic and aperiodic components of the RMBO network are distinctly coupled to respiration
In a final series of analyses, we aimed to disentangle the dynamics of respiratory coupling to periodic (i.e., oscillatory) and aperiodic (i.e., non-oscillatory) components of neural activity. So far, no investigation of respiration-brain coupling has directly considered either phase-locked changes of E:I balance or its distinction from oscillatory modulations. To target this question, we computed source-localised time series from the MEG data (mapped onto $n = 230$ parcels taken from the HCP atlas[21]) and defined regions of interest according to the original RMBO network publication[2]. For a total of $n = 10$ ROIs (consisting of $n = 23$ HCP parcels), we extracted respiration phase-locked 1/f slope as well as periodic spectra with the 1/f characteristic removed. For the computation of parcel-specific courses of aperiodic slope over respiratory phase, we followed the sensor-level approach outlined above. The periodic component was defined as the accumulated power of oscillatory peaks (between 1 and 40 Hz) after the 1/f slope was removed from the respective periodic spectrum. In keeping with our sensor-level analyses, we determined significance of respiration phase-locked changes in aperiodic slope and oscillatory power by means of LMEMs within each parcel of interest. Again, each parcel's empirical vector norm of LMEM beta weights for respiratory sine and cosine was assessed relative to $k = 5000$ null beta weights from LMEMs computed on surrogate distributions of slope and power over respiratory phases.

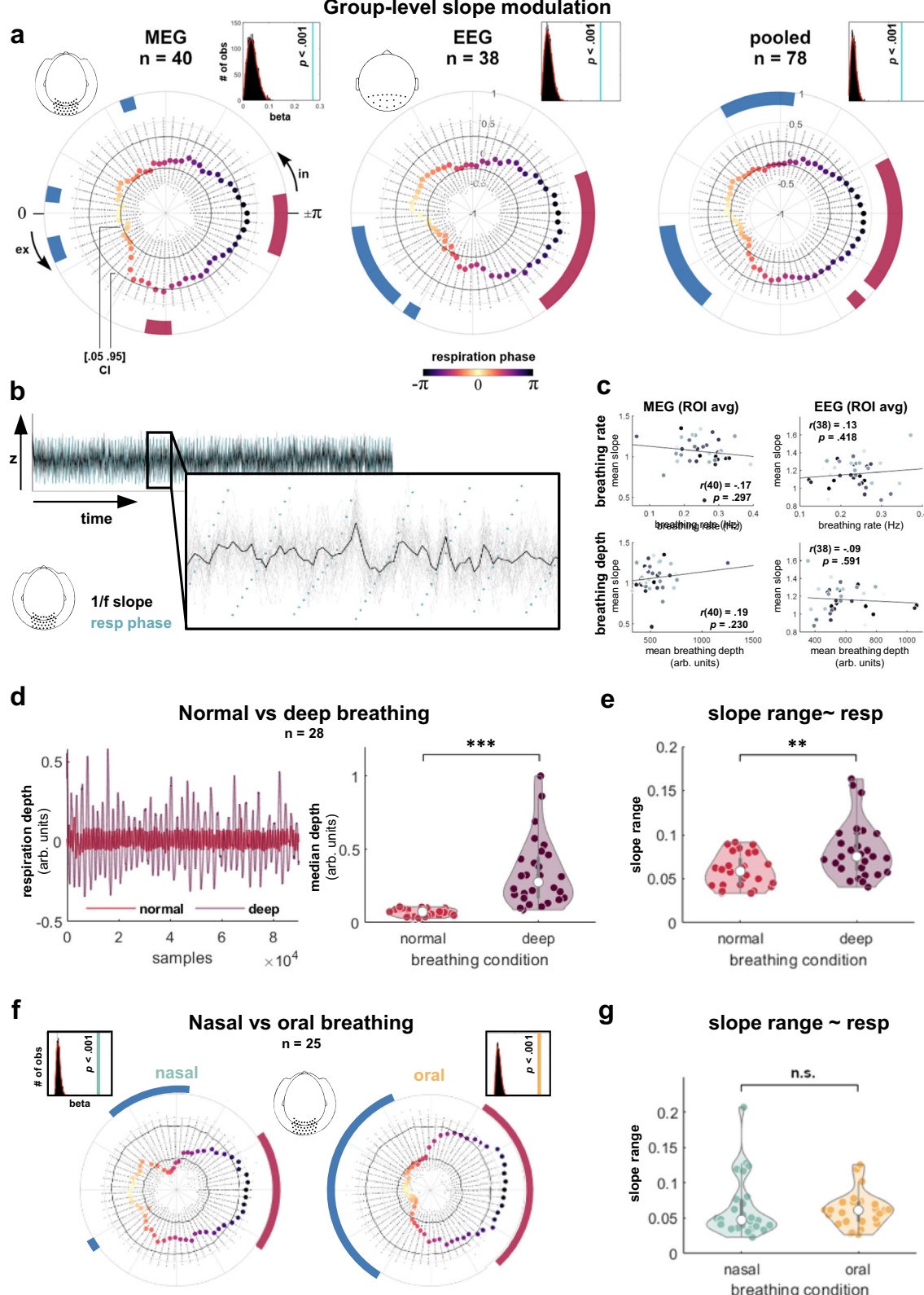

**Group-level slope modulation**

Of the 23 parcels of interest, 18 parcels showed a significant association between respiration and 1/f slope. Significant phase-locked modulations of oscillatory power (within frequencies from 1–40 Hz) were found for 19 parcels (see Fig. 4a). Overall, all 23 parcels showed either significant non-uniformity of slope or accumulated power. Next, we investigated differences in temporal dynamics between respective slope and power changes within each parcel. Specifically, we extracted the circular means of each parcel's slope and power courses (binned into $n = 60$ overlapping respiratory phase bins) to compute parcel-wise phase differences between the two courses: In case slope and power were similarly modulated by respiration, the mean phase difference across parcels would be (close to) zero. As shown in the polar histogram in Fig. 4b, this was not the case ($U^2 = 1.26$, $p < .001$, 5000 permutations). Next, we tested whether slope and power courses of each

**Fig. 2 | Group-level respiration-locked modulation of parieto-occipital 1/f slope. a** Respiration phase was extracted for $n = 60$ overlapping phase bins. For each data set, we show combined LMEM beta weights for the phase vector norm (light blue) against $k = 5000$ null betas (top right corners). Coloured bold dots show respiration phase-dependent parieto-occipital 1/f slope. Radial scatter plots indicate null distributions of $k = 5000$ bin-wise group-level mean exponent values. Solid black lines indicate the 5th and 95th percentile of each bin's null distribution. In the MEG sample, 1/f slope was flattened during inspiration (around −70°) and around the inspiration-to-expiration transition (−9° and 15° to 21°; blue markings). 1/f slope was steeper during expiration (82° to 88°) and around the expiration-to-inspiration transition (−174° to 162°; red markings). The EEG sample showed flatter slope after the inspiration-to-expiration transition (9° to 46° and around 58°) and steeper slope around the expiration-to-inspiration transition (−162° to 131°). In = inspiration, ex = expiration. **b** Overlay of ROI-average 1/f slope estimates (black) and respiratory phase (green) for a single MEG participant. Grey lines show single-

sensor slope estimates. **c** Neither breathing rate (top) nor depth (bottom) were significantly correlated with steepness of the aperiodic component in MEG (left) or EEG (right; two-sided $t$-test against zero). **d** Respiratory trace of a single participant (left) and median group-level breathing depths (right) during normal and deep breathing (re-analysed from[3]). Participants were breathing deeper during the deep (vs normal) breathing condition ($z(27) = 6.19$, $p < .001$, two-sided Wilcoxon test for $n = 28$ participants). **e** Individual ranges of slope-respiration phase were consistently greater for deep (compared to normal) breathing ($z(27) = 3.02$, $p = .003$, two-sided Wilcoxon test). **f** Comparison of parieto-occipital 1/f slope-respiration phase during nasal (left) and oral breathing (right) from a follow-up control study (see Methods). LMEMs confirmed a significant influence of respiratory phase on 1/f slope for both breathing conditions (see histogram insets). **g** Individual ranges of slope-respiration did not differ between conditions ($z(25) = 0.80$, $p = .426$, two-sided Wilcoxon test for $n = 26$ participants).

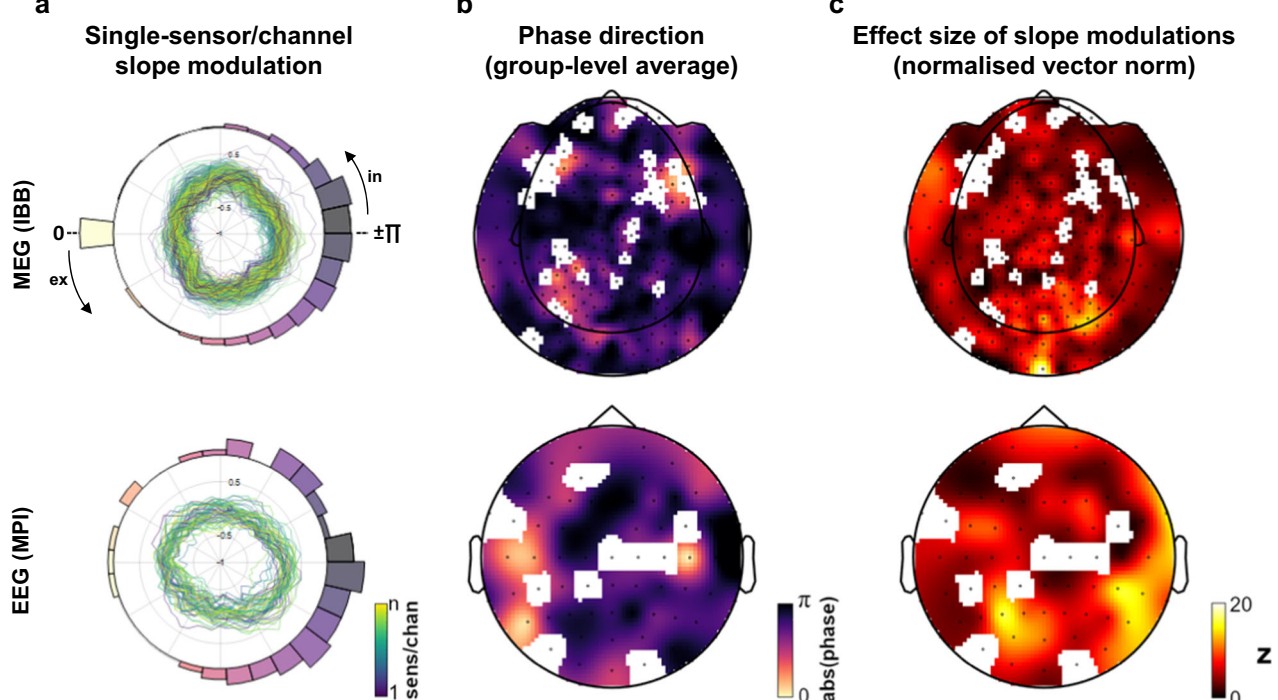

**a**
**Single-sensor/channel slope modulation**

**b**
**Phase direction (group-level average)**

**c**
**Effect size of slope modulations (normalised vector norm)**

MEG (IBB)

EEG (MPI)

**Fig. 3 | Single-sensor (MEG) and channel (EEG) modulation of 1/f slope. a** Polar plots shows group-level averages of normalised single-sensor/channel slope modulation courses over the respiration cycle. LMEM revealed 1/f slope over respiration to significantly deviate from a uniform distribution in $n = 240$ out of 275 MEG sensors and $n = 53$ out of 62 EEG channels. Only significant MEG sensors (top) and EEG channels (bottom) are shown, colour coding illustrates sensors and channels 1 to n. Polar histograms illustrate circular mean directions of single-sensor/channel slope distributions (colour-coded according to respiration phase). **b** Topographic

representations of significant modulations and their phase direction for MEG sensors (top) and EEG channels (bottom). **c** Population statistics quantifying the effect size of slope modulations seen in (**b**). For each sensor/channel, we show the vector norm of LMEM beta weights for sine and cosine of the respiratory phase (akin to a harmonic regression approach). These vector norms were normalised using the mean and standard deviation of the null distribution and are thus given in $z$ values (see Methods for details). In = inspiration, ex = expiration.

parcel were drawn from the same distribution. For all parcels, Watson's $U^2$ permutation tests confirmed significant differences between the underlying distributions (all $p < .001$), providing further evidence for differential modulatory dynamics of slope and power courses within the RMBO network. For both slope and accumulated power, Fig. 4c shows ROI-wise distributions of subject-level $U^2$ statistics for each parcel as an estimate of respective non-uniformity (centre panel). Exemplary comparisons between ROIs are shown for slope-respiration (left panel) and power spectra ~ respiration (right panel).

Finally, single-band modulations of 1/f-removed theta (4–8 Hz), alpha (8–13 Hz), and beta power (13–30 Hz) within the RMBO network are shown in Fig. 5. Out of 23 parcels, band-specific LMEMs revealed that theta power was significantly modulated in 17 parcels, alpha power in 22 parcels, and beta power in 21 parcels (see Fig. 5a and

Methods for details). While band-specific oscillatory modulations were not the main focus of this paper, we provide this characterisation to motivate hypothesis generation for future work.

## Discussion

Our present findings highlight the role of respiration as a physiological modulator of neural signalling. We provide evidence that fluctuations of aperiodic brain activity are phase-locked to the respiratory cycle which strongly suggests that spontaneous state shifts of excitation-inhibition balance are at least partly influenced by peripheral bodily signals. Moreover, differential temporal dynamics in their coupling to non-oscillatory and oscillatory activity raise the question whether there is a functional distinction in the way each component is related to respiration.

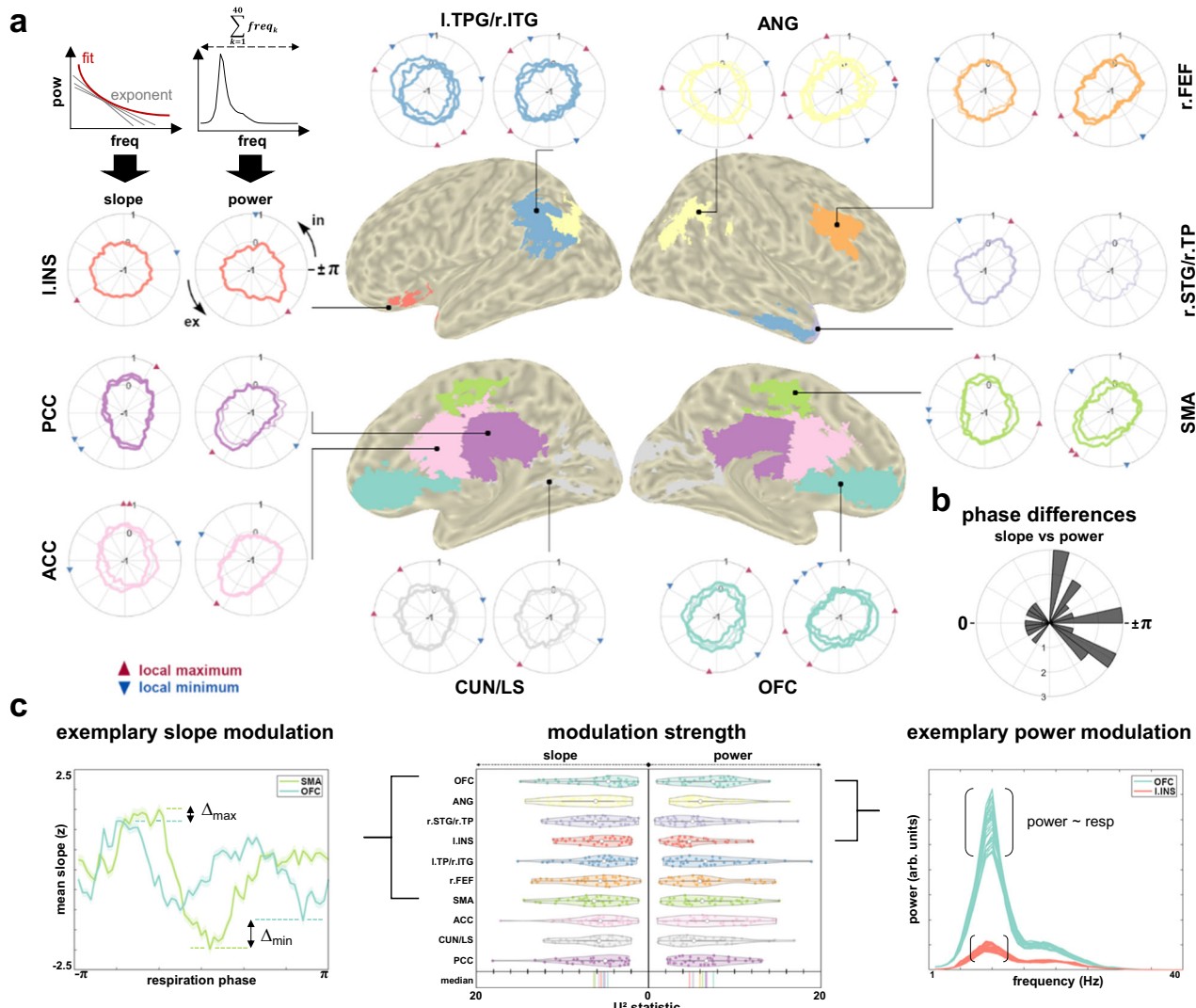

**Fig. 4 | Source-level ROI modulation of 1/f slope and accumulated oscillatory power in the MEG. a** For each cortical node of the RMBO network[2], we show the group-level average courses of 1/f slope (left polar plots) and accumulated spectral power after removal of the aperiodic component (1–40 Hz; right polar plots) over the respiratory cycle. For ROIs comprising more than one parcel from the HCP atlas, slope and power courses are superimposed for each individual parcel. In case the respective LMEM did not indicate significant association with either slope or power within any given parcel, the corresponding time series are drawn with reduced opacity. For significantly associated parcels only, red and blue triangles indicate respective maxima and minima over the respiratory cycle (see legend). In = inspiration, ex = expiration. **b** Polar histogram shows phase differences between the circular means of slope and power courses across parcels. The clear majority of non-zero phase differences were corroborated by Watson's $U^2$ statistics indicating distinct temporal dynamics of slope and power modulations within each parcel (see main text). **c** Centre: For each ROI, we show violin plots of $n = 40$ single-participant $U^2$ statistics (slope/power vs uniform distribution, respectively). Higher $U^2$ statistics indicate stronger non-uniformity. Left: Exemplary slope modulations of SMA and OFC (M ± SEM) normalised across respiration phase for illustration purposes. SMA was more strongly non-uniform over all (see centre), which is visible from both stronger maxima ($\Delta$max) and minima ($\Delta$min) compared to OFC. Right: Exemplary power modulations of OFC and l. INS. For both ROIs, we superimpose power spectra (with 1/f removed) from all 60 phase bins. Overall, power varied more strongly with respiratory phase within OFC compared to l.INS (see centre). TPJ = temporo-parietal junction, ITG/STG = inferior/superior temporal gyrus, ANG = angular gyrus, FEF = frontal eye-field, TP = temporal pole, SMA = supplementary motor area, OFC = orbitofrontal cortex, CUN = cuneus, LS = lingual sulcus, ACC/PCC = anterior/posterior cingulate cortex, INS = insular cortex.

While the mechanisms underlying the 1/f characteristic of neural power spectra have yet to be fully understood[22], crossmodal evidence has highlighted its functional significance in several domains. Most prominently, the steepness of the aperiodic slope (i.e., its exponent) has been shown to reflect the balance between excitatory (E) and inhibitory (I) synaptic currents in computational[16], pharmaceutical[15], as well as optogenetic work[23]: The larger the exponent (i.e., steeper power spectral slope), the stronger the inhibitory influence and vice versa[24]. In general, constant E:I balancing is essential to maintain neural homoeostasis[25] so that the excitability within a particular neural array remains at a critical state[26]. Facilitated by novel methodological advances, our present findings of respiration-locked changes in E:I balance thus illuminate a previously inaccessible aspect of what is emerging as a recurrent motif in respiration-brain coupling across species: Animal studies have shown excitability[27] as well as spike rates[12,13] to covary with the respiratory cycle, prompting non-invasive human work from our lab[9] in which we linked respiratory phase to excitability of visual cortices (measured as parieto-occipital alpha power) during low-level perception. Given that breathing is under volitional control, we adopted the theoretical perspective of predictive processing[28] to propose respiration as a means to actively align the sampling of external sensory information with transient internal states of increased excitability. This interpretation rests on the concept of active sensing[29] in rodents which recently gained further support from

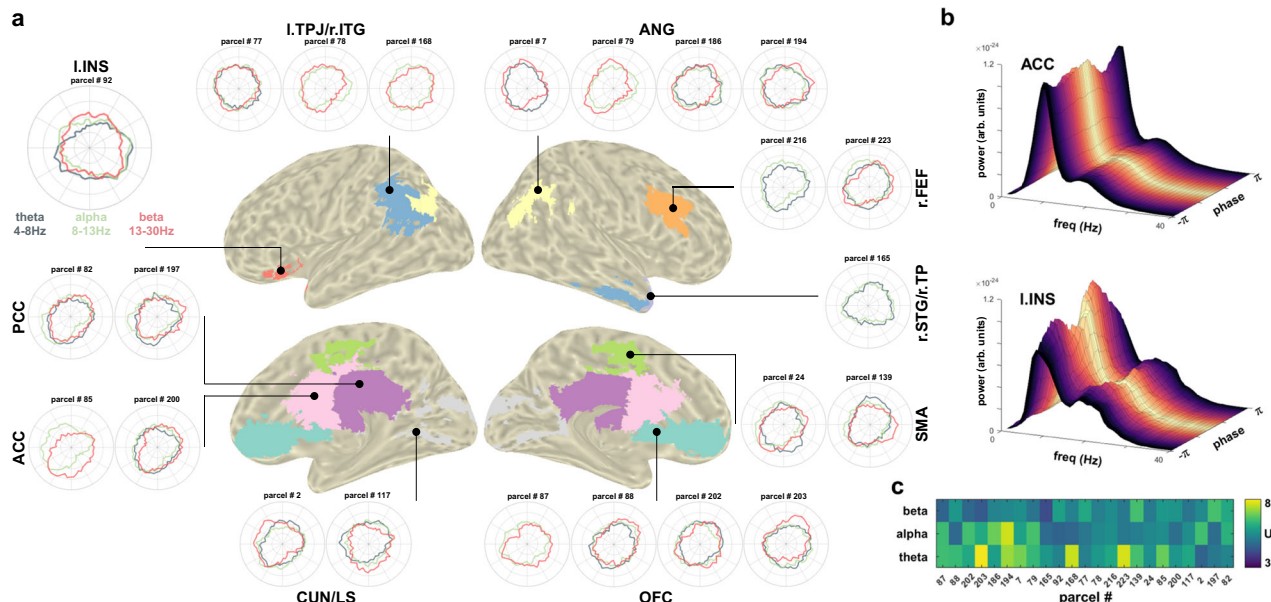

**Fig. 5 | Source-level ROI modulation of band-specific oscillatory power in the MEG. a** For each cortical node of the RMBO network, we show the group-level average courses of band-specific spectral power after removal of the aperiodic component over the respiratory cycle. For ROIs comprising more than one parcel from the HCP atlas, power courses are shown for each individual parcel. In case the band-specific LMEM did not indicate significant modulation of the periodic component within any given parcel, the corresponding power series are not shown. **b** Exemplary ROI-average spectra of anterior cingulate (top) and left insula (bottom), colour-coded according to respiratory phase. **c** Group-level median $U^2$ statistics of each parcel testing single-band power - respiration phase against a uniform distribution. Higher values indicate stronger deviation from uniform distribution.

a comprehensive demonstration by Karalis and Sirota[30]: In a series of elaborate experiments, they found neocortical up and down states during sleep and quiet rest to be modulated by respiratory phase, strongly suggesting that breathing influences (sub-)cortical excitability in the sense of a pacemaker aligning neural network dynamics. While the spontaneous aperiodic shifts of E:I balance in our paradigm are not to be equated to the reported up and down states in quiescent animals, we propose that the shared observation of respiration phase-locked state changes bears valuable insight into human respiration-brain coupling.

Investigating the respiration phase-locked changes in aperiodic vs periodic activity inevitably entails the more general distinction of 1/f and genuine oscillations per se and how previous works have addressed one or the other. To reiterate, neural oscillations are narrowband, rhythmic components embedded within a background of broadband, aperiodic activity which itself does not require any underlying rhythmicity[31,32]. So far, studies of breath-brain interactions have neglected aperiodic activity entirely, most likely due to the fact that one proposed mechanism of respiration-brain coupling critically relies on oscillatory entrainment to the breathing rhythm: Here, the nasal airstream entrains neural activity within the olfactory tract via mechanoreceptors, after which the phase of this infraslow rhythm is coupled to the amplitude of faster oscillations[12,33]. This propagation through phase-amplitude coupling[34] has largely been quantified by measures of coupling strength such as the modulation index (MI[35]), a method that – despite its benefits - prohibits any conclusion regarding non-oscillatory modulations. In the present study, we consistently observed steepest 1/f slopes around ± π, with strongest effects localised over posterior sensors, complementing previous reports of a spatial gradient of aperiodic dynamics[18,36]. Such a posterior-to-anterior gradient conceivably increases parieto-occipital SNR, which could facilitate the detection of respiration phase-locked changes. As for the pattern of preferred respiratory phase around ± π, He and colleagues[37] demonstrated a rich temporal organisation of aperiodic (or 'scale-free') brain activity in ECoG recordings from epilepsy patients: Extending the concept of hierarchically nested oscillations[38], the

authors showed nested frequencies in aperiodic activity during quiet wakefulness; in other words, coupling of low-frequency phase to high-frequency amplitude of 1/f slope. Intriguingly, the preferred phase clustered around phases 0 and ± π, i.e., peaks and troughs of low-frequency fluctuations, closely resembling present findings of flatter slopes (i.e., increased excitation) towards the end of inspiration as well as steeper slopes (i.e., more inhibition) towards the end of expiration (see Figs. 2 and 3). These aperiodic shifts of E:I state further corroborate previous findings of increased excitability during inspiration[9] and improved cognitive performance for stimuli presented during the inspiratory phase[6,7]. If the proposed functional mechanisms of respiration-brain coupling were indeed to be extended to include aperiodic brain activity, respiratory modulations of 1/f slope should be flexibly adapted during cognitive or perceptual tasks to facilitate performance. A study by Waschke and colleagues[15] recently provided convincing evidence that the aperiodic component itself not only captured task-relevant attentional changes in E:I balance, but also covaried with individual behavioural performance. Critically, the authors used anaesthetics to distinguish between oscillatory (i.e., increased band-specific power) and non-oscillatory, aperiodic changes in EEG spectra. In a similar vein, Donoghue et al.[18] stressed the importance of disentangling periodic and aperiodic changes in neural activity in order to recognise their potentially distinct contributions to observed effects. We argue that the same is true in order to more comprehensively describe the physiological underpinnings of respiration-brain coupling.

In the present data, the preferred phase differences between source-level periodic and aperiodic modulations within the RMBO network show at least partly differential dynamics between both components, suggesting a more complex coupling of neural signalling to the respiratory rhythm than previously assumed. Conceivably, changes in periodic and aperiodic activity could reflect respiration-locked neural modulations arising from two (not necessarily) independent mechanisms: While the respiratory rhythm itself is generated in the brain stem[39], oscillatory changes on the one hand are assumed to be induced by mechanosensory feedback stimulation from the nasal

airstream during inspiration and expiration (see above). Convincing evidence for this hypothesis comes from studies showing that both neural and behavioural modulations dissipate when there is no nasal airflow stimulating olfactory mechanoreceptors, i.e. during oral breathing[6,7] or in bulbectomised animals[12]. On the other hand, $CO_2$ levels continuously regulate blood flow and respiration rate in a feedforward fashion, which in turn determine changes in $CO_2$ levels[40]. Moreover, $CO_2$ is inversely related to tissue acidity (pH), which has been linked to neural excitability[41,42]. Lowered pH (due to increased $CO_2$) causes an increase in extracellular adenosine[43], a neuromodulator gating synaptic transmission. Consequently, Dulla and colleagues[27] concluded that $CO_2$-induced changes in neural excitability are caused by pH-dependent modulation of adenosine and ATP levels (also see[44]).

Since $CO_2$ levels do not differ between nasal and oral breathing, the two mechanisms are clearly functionally distinct from one another: $CO_2$ fluctuations alone cannot account for neural or behavioural differences between nasal and oral breathing. Hence, there are (at least) two different ways in which neural activity could be coupled to the act of breathing. It may be the case that the differential coupling of respiration to periodic and aperiodic activity reflects these distinct mechanisms. However, the evidence so far (including present data from the control study) neither precludes the possibility that both types of modulation could be rooted in the same underlying mechanism. Therefore, critical future work is needed to further elucidate the functional and anatomical underpinnings of respiratory modulations in brain dynamics.

The present findings open multiple avenues for future research, of which we want to briefly sketch three main directions. First, mechanistic advances could be made in a replicative study which includes a contrast of nasal vs oral respiration. The absence of phase-amplitude coupling driven by nasal airstreams during oral breathing could conceivably reveal to what extent aperiodic changes rely on and/or interact with oscillatory changes. While we speculate that respective changes in 1/f slope and oscillatory power could point to distinct modulatory pathways, targeted follow-up studies with sufficient statistical power are of critical importance to extend our initial results. Second, as outlined above, it would further be highly instructive to complement existing evidence of task-related 1/f slope changes[23,45] by investigating their link to respiration in behavioural cognitive or perceptual paradigms. Timed breathing (in healthy participants or ventilated patients) or contexts involving different brain states, e.g. resting-state vs task, during attentional or arousal manipulation (including hyperventilation), or during sleep stages, could potentially unravel the link between breathing and fluctuations of (non-)oscillatory brain activity.

Finally, a third line of research should focus on potential translational applications of respiration-brain coupling. While others have made a convincing case for studying respiratory involvement in clinical contexts before (for a recent review, see[46]), the results presented here demonstrate a link between bodily signals and aperiodic neural activity. Importantly, changes in 1/f slope itself - an indicator of E:I imbalance - have been implicated in neurological and psychiatric disorders like Alzheimer's disease[47], schizophrenia[48], autism spectrum disorder[49], and epilepsy[50]. Linking these known neural alterations to peripheral signals could provide substantial insight into body-brain interactions in health and disease.

## Methods

### Participants and data acquisition (Münster)
Forty right-handed volunteers (21 female, age $25.1 \pm 2.7$ y [M ± SD]) participated in the study. All participants reported having no respiratory or neurological disease and gave written informed consent prior to all experimental procedures. The study was approved by the local ethics committee of the University of Münster (approval ID 2018-068-f-S). All participants provided written informed consent and received financial

compensation for their participation in the study. Participants were seated upright in a magnetically shielded room while we simultaneously recorded 5 min of MEG and respiratory data. MEG data was acquired using a 275 channel whole-head system (OMEGA 275, VSM Medtech Ltd., Vancouver, Canada) at a sampling frequency of 600 Hz. During recording, participants were to keep their eyes on a fixation cross centred on a projector screen placed in front of them. To minimise head movement, participants' heads were stabilised with cotton pads inside the MEG helmet. Participants were instructed to breathe naturally through their nose while the respiratory signal was measured as thoracic circumference by means of a respiration belt transducer (BIOPAC Systems, Goleta, USA) placed around their chest. Continuous monitoring via video ensured participants were breathing through their nose instead of their mouth. Individual respiration time courses were visually inspected for irregular breathing patterns such as breath holds or unusual breathing frequencies, but no such artefacts were detected.

### Participants and data acquisition (Leipzig)
Thirty-eight healthy volunteers (18 female, age $27.1 \pm 4.0$ y [M ± SD]) were recruited from the database of the Max Planck Institute for Human Cognitive and Brain Sciences, Leipzig, Germany. Participants reported no history or current neurological or psychological condition. The study was approved by the Ethical Committee of the University of Leipzig's Medical Faculty (No. 462-15-28082020). All participants provided written informed consent and received financial compensation for their participation in the study. Participants were seated upright in an EEG booth while 5 min of EEG and respiration were recorded. EEG was recorded from 62 scalp positions distributed over both hemispheres according to the international 10–10 system, using a commercial EEG acquisition system (ActiCap Snap, BrainAmp; Brain Products). The mid-frontal electrode (FCz) was used as the reference and a mid-frontal electrode placed on the middle part of the forehead (between FP1 and FP2) as ground. Electrode TP9 was used to measure ECG and electrode TP10 captured eye movements. Electrode impedance was kept below 10 kΩ for all channels. EEG was sampled with a rate of 2.5 kHz and online bandpass-filtered between 0.015 and 1000 Hz. Participants were instructed to keep their eyes open and fixate a cross on the screen in front of them to avoid excess eye movements. Participants were instructed to breathe naturally while respiration was measured with a respiration belt transducer (BIOPAC Systems, Goleta, USA) placed around the chest.

### Participants and data acquisition (control MEG study for deep vs normal breathing)
The MEG sample for the comparison of deep vs normal breathing has previously been published elsewhere[3]. Twenty-eight volunteers (14 female, age $24.8 \pm 2.9$ y [M ± SD]) participated in the study conducted at the Institute for Biomagnetism and Biosignal Analysis in Münster. All participants reported having no respiratory or neurological disease and gave written informed consent prior to all experimental procedures. The study was approved by the local ethics committee of the University of Münster (approval ID 2018-068-f-S). MEG recording parameters and procedures were identical to the original recordings described above. Data were acquired in two 5-min runs with a short intermediate break (determined by the participants). Within each run, participants were instructed to either breathe normally or voluntarily deeply (with their mouth closed) while maintaining their normal respiration frequency, i.e. not to breathe more slowly during blocks of deep breathing (see[3]). Continuous monitoring via video ensured participants did indeed keep their mouth closed throughout the recording. The order of deep and normal breathing was counterbalanced across participants. For both runs, we recorded the respiratory signal as thoracic circumference by means of a respiration belt transducer (BIOPAC Systems, Goleta, USA) placed around the participant's chest. Individual respiration time courses were visually inspected for

irregular breathing patterns such as breath holds or unusual breathing frequencies, but no such artefacts were detected.

## Participants and data acquisition (control MEG study for nasal vs oral breathing)

Twenty-five right-handed volunteers (10 female, age $26.3 \pm 3.3$ y [M ± SD]) participated in the control MEG study conducted at the Institute for Biomagnetism and Biosignal Analysis in Münster. All participants reported having no respiratory or neurological disease and gave written informed consent prior to all experimental procedures. The study was approved by the local ethics committee of the University of Münster (approval ID 2021-785-f-S). MEG recording parameters and procedures were identical to the original recordings described above. For each participant, we recorded two 6-min runs of resting state activity: In one run, participants were instructed to breathe naturally through their nose. Continuous monitoring via video ensured participants did indeed keep their mouth closed throughout the recording. In the other run, participants were instructed to breathe through their mouth while wearing a nose clip to prevent nasal breathing. The order of nasal and oral breathing was counterbalanced across participants. Again, we recorded the respiratory signal as thoracic circumference by means of a respiration belt transducer (BIOPAC Systems, Goleta, USA) placed around the participant's chest. Individual respiration time courses were visually inspected for irregular breathing patterns such as breath holds or unusual breathing frequencies, but no such artefacts were detected.

## MRI acquisition and co-registration (Münster)

For MEG source localisation, we obtained high-resolution structural magnetic resonance imaging (MRI) scans in a 3 T Magnetom Prisma scanner (Siemens, Erlangen, Germany). Anatomical images were acquired using a standard Siemens 3D T1-weighted whole-brain MPRAGE imaging sequence ($1 \times 1 \times 1$ mm voxel size, TR = 2130 ms, TE = 3.51 ms, $256 \times 256$ mm field of view, 192 sagittal slices). MRI measurement was conducted in supine position to reduce head movements, and gadolinium markers were placed at the nasion as well as left and right distal outer ear canal positions for landmark-based co-registration of MEG and MRI coordinate systems. Data preprocessing was performed using Fieldtrip[51] running in MATLAB R2021a (The Mathworks, Natick, USA). Individual raw MEG data were visually inspected for jump artefacts and bad channels, but neither were detected. Both MEG and respiration data were resampled to 300 Hz prior to further analyses.

Co-registration of structural T1 MRIs to the MEG coordinate system was done for each participant by initial identification of three anatomical landmarks (nasion, left and right pre-auricular points) in their individual MRI. Using the implemented segmentation algorithms in Fieldtrip and SPM12, individual head models were constructed from anatomical MRIs. A solution of the forward model was computed using the realistically-shaped single-shell volume conductor model[52] with a 5 mm grid defined in the Human Connectome Project (HCP) template brain[21] after linear transformation to the individual MRI.

## Respiratory preprocessing (both labs)

To obtain continuous respiration phase angles, we used Matlab's findpeaks function (with minimal peak prominence set to 1) to identify time points of peak inspiration (peaks) and peak expiration (troughs) in the normalised respiration time course. Phase angles were linearly interpolated from trough to peak ($-\pi$ to 0) and peak-to trough (0 to $\pi$) in order to yield respiration cycles centred around peak inspiration (i.e., phase 0).

In order to assess the potential influence of individual breathing parameters on aperiodic fluctuations, we computed breathing rates and depths for each participant. Breathing rates were extracted as the mean distance between inspiratory peaks as defined by the peak

detection algorithm described above. Breathing depths were computed as the integral of the individual respiration time series, normalised by the number of breathing cycles during the recording.

## Head movement correction (Münster)

In order to rule out head movement as a potential confound in our analyses, we used a correction method established by Stolk and colleagues[53]. This method uses the accurate online head movement tracking that is performed by our acquisition system during MEG recordings (as previously used in[2,3]). This leads to six continuous signals (temporally aligned with the MEG signal) that represent the x, y, and z coordinates of the head centre ($H_x$, $H_y$, $H_z$) and the three rotation angles ($H_\psi$, $H_\theta$, $H_\varphi$) that together fully describe head movement. We constructed a regression model comprising these six 'raw' signals as well as their derivatives and, from these 12 signals, the first-, second-, and third-order non-linear regressors to compute a total of 36 head movement-related regression weights (using a third-order polynomial fit to remove slow drifts). This regression analysis was performed on the power spectra of single-sensor and single-voxel time courses, respectively, removing signal components that can be explained by translation or rotation of the head with respect to the MEG sensors.

## Respiration-locked computations of 1/f slope and oscillatory power

Our first aim was to further investigate excitability changes over posterior cortices we previously reported[9]. To this end, we defined parieto-occipital regions of interest for movement-corrected MEG ($k = 41$ sensors) as well as EEG data ($k = 17$ channels). Single-sensor/ channel time series within these ROIs were entered into the SPRiNT algorithm[19] with default parameter settings and subsequently averaged. In short, SPRiNT is based on the specparam algorithm[18] and uses a short-time Fourier transform (frequency range 1–40 Hz) to compute aperiodic and periodic components of neural time series within a moving window (width of 1 s, 75% overlap between two neighbouring windows). Due to the slow nature of the respiratory signal, we did not average between neighbouring windows at this point (see below for details). SPRiNT thus yielded time series of both aperiodic (i.e., 1/f exponent) and periodic signals (i.e., oscillatory power) with a temporal resolution of 250 ms and a frequency resolution of 1 Hz. In order to relate these time series to the respiratory signal, we extracted respiratory phase at all time points for which slope and power were fitted (i.e., the centres of each moving window). Following previous work[2,9], we then partitioned the entire respiratory cycle ($-\pi$ to $\pi$) into $n = 60$ equidistant, overlapping phase bins. Moving along the respiration cycle in increments of $\Delta\omega = \pi/30$, we collected all SPRiNT outputs (i.e., slope fits and periodic spectra) computed at a respiration angle of $\omega \pm \pi/10$. At this point, we computed individual bin-wise averages of 1/f slope and periodic spectra, yielding quasi-continuous 'phase courses' of aperiodic and periodic neural signals for each participant (see Supplementary Figs. S1 and S2 for bin-wise event numbers). Supplementary Figs. S3 and S4 show phase-locked power spectra over parieto-occipital MEG sensors and EEG channels, respectively. For the control MEG studies, we separately applied this pipeline to the MEG data recorded during deep/normal and nasal/oral breathing, respectively. The methodological approach is illustrated in Fig. 1.

For the source-level MEG analyses shown in Figs. 4 and 5, SPRiNT computed 1/f slope fits and periodic spectra based on the source-localised time series of individual parcels from the HCP atlas.

## Statistical analysis of sensor-level 1/f slope modulation

As a first analysis of respiration phase-locked changes in 1/f slope, we computed the following linear mixed-effects model (LMEM):

$$SE_j = \beta_0 + (S_{1,j} + \beta_1) * \sin_{resp} + (S_{2,j} = \beta_2) * \cos_{resp} + e_j \quad (1)$$

For each participant j, the model predicted individual slope (resulting from SPRiNT computations, see above) as a combination of the intercept ($\beta_0$), fixed effects of respiratory sine and cosine ($\beta_1$, $\beta_2$), and an error term ($e_j \sim N(0, \sigma2)$). In line with previous work[9], resulting beta weights for sine and cosine of the respiratory signal were combined in a respiratory phase vector norm, i.e.

$$v = \sqrt{\beta_1^2 + \beta_2^2} \qquad (2)$$

To test whether 1/f was significantly modulated over the ROI sensors, we computed $k = 5000$ random iterations of the LMEM by shuffling bin-wise 1/f slope fits on the individual level. This way, any meaningful relations between 1/f slope and respiratory phase were removed, thereby specifically testing the hypothesis that 1/f slope changes significantly with respiratory phase. For each of these iterations, we once again computed the phase vector norm from the resulting beta weights, yielding a random distribution of $k = 5000$ 'null vector norms'. Significance of the empirical vector norm was determined by its percentile relative to this null distribution. As shown in Figs. 2 and 3, for the sensor-level analyses, SPRiNT parameterisation was conducted on individual MEG sensors (Münster) and EEG channels (Leipzig). For the analysis of 1/f slope modulations within the parieto-occipital ROI, we computed mean SPRiNT estimates across the corresponding sensors or channels, respectively. LMEMs were then computed on ROI-average estimates for 1/f slope-respiration phase for each lab individually (Münster, Leipzig) as well as for the pooled data from both labs (see Fig. 2). For the whole-scalp analyses shown in Fig. 3, LMEMs were computed on single-sensor/channel SPRiNT estimates. Having established an over all significant influence of respiratory phase on 1/f slope with the LMEM approach described above, we next aimed to characterise when (i.e., at which respiratory phases) 1/f slope was significantly steeper or flatter. To this end, we implemented a permutation approach as follows: For each participant, we constructed a surrogate respiration time series using the iterated amplitude-adjusted Fourier transform (IAAFT[54]). In contrast to shuffling the respiration time series, this iterative procedure preserves the temporal autocorrelation of the signal, which is critical for constructing a fitting null distribution for permutation testing. From these IAAFT-transformed respiration time series, we extracted the surrogate respiratory phase values corresponding to each slope estimation from SPRiNT. In keeping with the approach above, we finally binned all SPRiNT outputs into $n = 60$ equidistant, overlapping phase bins covering one entire respiratory cycle (-π to π) and computed the bin-wise average slope fit to yield a 'null time series' of 1/f slope over surrogate respiration phase. For each participant, this procedure was repeated 5000 times and resulted in a null distribution of 5000 surrogate 1/f slope estimates x 60 phase bins. We then computed the group-level average null distribution of 1/f slope for each of the 60 phase bins which allowed us to extract the percentiles of empirical bin-wise group means relative to these null distributions. As illustrated in Fig. 2, empirical group-level means were deemed significant if they exceeded the 95th percentile (or fell below the 5th percentile) of the null distribution.

For single-sensor slope analyses (Fig. 3) as well as single-parcel analyses of 1/f slope and periodic spectra (Fig. 4), we used the circstat toolbox for Matlab[55] to compute the circular means of group-level average slope (and power) courses over the respiratory cycle. In the absence of meaningful slope modulation by respiratory phase, a polar illustration of the binned slope estimates would be a uniform circle. In contrast, any phase-locked slope changes would be represented as modulations or peaks within the polar representation. Albeit not for all distributions, the circular mean provides at least some indication with regard to the direction of such modulations (see below for alternative measures). Therefore, we used it to characterise the consistency of slope modulations across the whole scalp both within MEG/EEG data as well as across both modalities and labs (see Fig. 3).

In addition to potential respiration-related modulations of slope courses, we quantified the strength of these phase-locked modulations. To this end, we computed z scores from sensor-specific LMEM beta weights of the respiratory phase vector norm (see above for details) by subtracting the mean of its null distribution and dividing by the corresponding standard deviation:

$$z = \frac{v - \mu_{v(null)}}{\sigma_{v(null)}} \qquad (3)$$

with v being the vector norm of beta weights for respiratory sine and cosine (see Eq. 2). This way, empirical LMEM beta weights for the respiratory phase vector are given in units of standard deviation relative to their null distribution.

## Source reconstruction

Source reconstruction was performed using the linearly constrained minimum variance (LCMV) beamformer approach[56] with the lambda regularisation parameter set to 5%. This approach estimates a spatial filter for each location of the 5 mm grid along the direction yielding maximum power. A single broadband LCMV beamformer was used to estimate voxel-level activities across all frequencies.

## ROI-based analyses of source-level 1/f slope and oscillatory power

For the investigation of source-level modulations of 1/f slope and periodic spectra, we first extracted neural time series from a total of $n = 230$ cortical parcels from the HCP atlas[21]. In order to ground our analyses in previous findings, we then focussed our analyses on those parcels representing the cortical nodes of the RMBO network[2]. This network has been demonstrated to show respiration-modulated brain oscillations (RMBOs) during resting state, which is why it was particularly well-suited to investigate potentially distinct dynamics of periodic and aperiodic modulations. As some RMBO nodes comprised bilateral or wider-spread anatomical sites, a total of $N = 10$ ROIs consisting of $n = 23$ parcels were used for source-level analyses (see Fig. 4). For the average time series within these parcels, we applied the SPRiNT analysis described above to yield both 1/f slope and oscillatory power across the respiratory cycle. For the latter, SPRiNT computed periodic spectra between 1 and 40 Hz with the aperiodic component removed, i.e., solely the power of the periodic neural signal. Just as with the 1/f exponents, we sorted all periodic spectra with respect to the corresponding respiratory phase to yield individual bin-average periodic spectra for $n = 60$ overlapping phase bins. These spectra thus reflect, for each phase of the respiratory cycle, the extent of 'true' oscillatory activity remaining after the underlying 1/f characteristic is removed. In a first step, we quantified the amount of oscillatory activity by computing the accumulated sum over all frequencies (illustrated in Fig. 4). For frequency band-specific power courses, we then computed accumulated power for canonical bands separately (theta: 4–8 Hz, alpha: 8–13 Hz, beta: 13–30 Hz).

Next, we aimed to investigate potentially distinct respiration phase-locked dynamics of periodic and aperiodic modulations. Therefore, for each HCP parcel, we implemented Watson's $U^2$ tests with $k = 5000$ random permutations (watsons_$U^2$_perm_test function[57] for Matlab) to test the null hypothesis that the group-level average vectors of slope and power were drawn from the same distribution. We chose the $U^2$ test as an additional measure as it is sensitive towards multimodal distributions[58], which is not the case for the circular mean alone. For the sake of comparability, however, we also computed phase differences between each parcel's circular means of slope and power courses: If 1/f slope and oscillatory power were similarly coupled to the respiratory rhythm, their respective courses over respiratory phase would not differ and the phase difference between their circular means would be (close to) zero. In contrast, systematically

different distributions of slope and power would suggest differential dynamics in their respective coupling to respiratory phase.

Finally, we characterised the strength of slope and power modulations for each of the $n = 10$ RMBO ROIs by means of $U^2$ statistics. For each participant, we conducted Watson's $U^2$ tests of the (potentially multimodal) courses of 1/f slope and accumulated power within each ROI against a uniform distribution. Here, higher $U^2$ values indicate stronger non-uniformity within a given parcel, which allowed us to characterise RMBO ROIs by their respective group-level $U^2$ distributions (as shown in Fig. 4c).

### Reporting summary

Further information on research design is available in the Nature Portfolio Reporting Summary linked to this article.

## Data availability

The processed data generated in this study are publicly available from the Open Science Framework (https://osf.io/8nw9t/). Processed and re-analysed data from the control MEG study for deep vs normal breathing (see[3]) are publicly available from the same folder. While ethics protocols disallow publicly sharing the raw data of the present study as well as the control MEG study, these data may be shared upon request. To obtain the data, please contact the corresponding author, Daniel Kluger (daniel.kluger@wwu.de). Source data are provided with this paper.

## Code availability

All custom Matlab code to reproduce the central findings of this study are publicly available from the Open Science Framework (https://osf.io/8nw9t/).

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

## Acknowledgements

We would like to thank Karin Wilken, Ute Trompeter, and Hildegard Deitermann for their invaluable assistance during data collection in Münster. This work was supported by the Interdisciplinary Centre for Clinical Research (IZKF) of the medical faculty of Münster (Gro3/001/19, awarded to JG). DSK (KL-3580/1-1), AV (Research Training Group 2386), and JG (GR 2024/ 5-1) were further supported by the DFG.

## Author contributions

Conceptualisation, D.S.K.; Methodology, D.S.K., C.F., J.G.;Investigation, D.S.K., C.F.; Writing – Original Draft, D.S.K., C.F.; Writing – Review & Editing, D.S.K., C.F., O.A., N.C., A.V., J.G.; Visualisation, D.S.K.; Funding Acquisition, D.S.K., A.V., J.G.

## Funding

## Competing interests

The authors declare no competing interests.
