## [Peer Review File · Nature Communications]

REVIEWER COMMENTS

Reviewer #1 (Remarks to the Author):

The manuscript by Kluger et al. uses recent new methodology for parametrizing the aperiodic and periodic components of the spectrogram of human neurophysiological time series, to study whether these latter are related to the phase of the respiratory cycle. This is a significant question because the latter is related to the sensitivity of sensory perception, while some characteristics of the aperiodic component of the neurophysiological frequency spectrum have been proposed to be related to neural excitability. The temporal alignment between independent observations derived from human brain and body physiology opens meaningful new perspectives on perception and related behaviour, and would inform future studies of their trial-to-trial fluctuations.

The manuscript provides sufficient background motivation to the reported study. One strength is to report data pooled from two different sites, each providing distinct modalities (MEG vs. EEG), with a reasonable N of about 40 participants each.

The manuscript is overall well written, but there are several instances where the approach is unclearly reported, both in the text and the figures, and where the methodology needs to be revised or at least, better justified.

Taken together, I believe the manuscript has potential but also shortcomings. It may better fit a more specialized journal.

Major points:

- Several of the null models used for testing the linear mixed effect models resort to randomized individual respiration time series. It is crucial that the randomized versions of the respiration phase preserve the original signal's temporal autocorrelation, to avoid deflating the specificity of the resulting tests.
- MEG head movements have been rightfully corrected for and regressed out of the analyses. Yet, to ascertain that artefactual residuals did not contaminate the analyses, at least one control test should be provided in Supplementary Material substituting the respiration time series with the principal component (from PCA) of all recorded movement time series.
- The conclusions (p.6) that respiratory phase influence the aperiodic neural signals is incorrect. The linear relationship of the mixed effect model does not establish a causal influence, simply an association.
- The rationale for presenting the pooled data in addition to the two datasets analyzed separately needs to be clarified. What do we learn from pooling the two datasets?
- Some key aspects of Figure 2 are unclear. The "concentric columns" referred in the main text look like concentric dashed lines that are virtually identical across all phase bins. Further, the black lines supposed to show the 5th and 95th percentiles are not seen in the figure. Further, it is unclear what the inner dark circle represents in each polar plot. Also, contrarily to the text statements, the blue markings are in the early phase of expiration for EEG, not the late phase of inspiration.
- 'exp' is an ambiguous contraction as it could be interpreted as 'exponent' in the present context (this is a minor point).
- How to read Table 1 is entirely unclear, despite the caption. The layout needs to be clarified. As a more general comment, testing the phase disparity between the MEG and EEG datasets needs to be conducted in a quantitative fashion and discussed.
- P.8: This notion of shift is not well explained in the text main body. I suppose this represents a form of bias indicating around which respiratory phase the slope tends to be the strongest? The rationale and definition for this measure need to be entirely clear from

the text.

- Figure 3: The topographies look noisy and not evidently consistent between the MEG and EEG data, which questions the significance of the data interpretation. Also, the strength of the slope course is a notion that is not clearly explained. The colorscale of Panel a shows a sens/chan unit, which meaning is also unclear.

- Figure 4: Overall, the figure is very busy and hard to read. More importantly, the similarity of the shape of the slope and power polar plots is truly not convincing enough. This is emphasized by the only scattered occurrences of when such similarities are statistically significant along the unit circle.

Minor points:

- Figure 1 is a cartoon illustration of the approach, yet it is confusing. The illustration of the "aperiodic slope" in the center frame does not show a log-log plot of the aperiodic spectrum, and therefore not a straight line representation of the latter. Hence the notion of slope is meaningless here.

Symmetrically, the periodic spectrum shows the full parametrized spectrum, which includes the aperiodic component.

Therefore Figure 1 requires changes to make sure the figure didactically and accurately informs the reader about the approach.

I believe the peer-reviewed version of SPRiNT just published in eLife provides a spectrogram representation of the periodic and aperiodic parameter estimates. This may be considered to clarify their time-varying nature in this figure and maybe in the main body of results of the paper.

Reviewer #2 (Remarks to the Author):

Kluger and coworkers have analyzed the effects of respiration on rhythmic and non-rhythmic activity in the human neocortex. Using MEG- and EEG-recordings, respectively, from two cooperating groups, the authors show that the slope of the spectrogram ($\log(\text{power})$ vs. $\log(\text{frequency})$) varies along the breathing cycle in resting humans. Moreover, they find a differential respiratory modulation of the non-rhythmic versus the rhythmic components of the power spectrum. Based on recent work on aperiodic brain activity (esp. from B. Voytek and colleagues) they interpret their findings as evidence for respiration-dependent fluctuations in excitation-inhibition cycle. Furthermore, they interpret the different cycle-dependent modulation of oscillatory (periodic) versus non-oscillatory (aperiodic) activity as evidence for different underlying mechanisms.

The groups have long track records in human electrophysiology, and the paper adds to previous work along the same line (Kluger et al., eLife 2021). The question is certainly relevant and I share the author's impression that it has not been studied before - analyzing E/I-balance by using the $1/f$ -slope of spectrograms is a relatively new approach (Gao et al., 2017), and connecting this measure with the breathing cycle requires using new and advanced non-stationary analysis tools (Wilson et al., BioRxiv 2022). Thus, the present contribution makes a significant contribution to a new line of research focusing on non-oscillatory brain activity, a highly understudied field in network- and cognitive neurosciences.

Despite this very positive general impression, I have some conceptual and editorial issues:

1. Lack of interventional or correlative manipulations. At present, the study appears a bit like a highly skillful application of the new analysis routine, while the biological significance

of the findings does not become entirely clear. The electrophysiological observation seems to be robust: most brain regions are modulated by respiration; this applies to both for 1/f and cumulative oscillatory power; the phase-wise modulation differs between both constituents of electrographic activity. However, we learn little about the behavioral or cognitive relevance, state-dependence, modulation by altered breathing etc. Of course, the authors cannot test all these factors together with the first description. However, if the phenomena were sensitive to any intervention, confidence in the biological/cognitive function would increase and hypotheses regarding the mechanism would be substantiated. For example: is there any correlation with breathing parameters (frequency, depth, mouth versus nose)? Is the phenomenon altered (at least in pilot experiments) when subjects leave the default state and concentrate on a task, e.g. visual? Is there any correlation with arousal? These questions are just examples – the critical point is that any such experiment could bring the data from a 'mere' description (not meant as a pejorative term) to a level stimulating further studies. Note: the feeling that 'something is lacking' may partly result from the somehow unimpressive presentation of data (see my comment #3).

2. Interpretation. The observational character of the study prompts the authors to build mechanistic hypotheses, which is a perfectly valid approach. Two of these hypotheses seem problematic to this reviewer:

2.1 The respiration-dependence constitutes a somatic feedback signal. I agree that there is ample evidence for such somatic feedback (as quoted by the authors) but, at the same time, it is not clear that purely neurogenic mechanisms, starting from central pattern generators in the brainstem, do not play a role (see the quoted paper by Karalis and Sirota). This may even differ between brain regions or between aperiodic and periodic activity. Therefore, the feedback hypothesis should be put forward with some more caution, and the alternative mechanism(s) should be made more explicit. Ideally, changing from nasal to mouth respiration would be a simple and instructive experiment to add (see my comment #1).

2.2 The statement that the differences in modulation phase of aperiodic/periodic activity 'show at least partly differential dynamics' (discussion) or that these differences 'point towards a functional distinction...' are rather strong. Oscillatory activity is likely based on specific mechanisms which are distinct from those underlying non-oscillatory activity. If a common factor, let us say a modulatory afferent signal from primary olfactory regions, affects both patterns of activity, differences in the distribution of power versus phase may just reflect the different nature of the signal that has been analyzed. This would be a 'trivial' (though potentially important) difference, but from the point of respiration rooted in one identical mechanism. This discussion needs no be reflected extensively in the paper, but in my view statements about differences in mechanisms or biological/cognitive consequences should be toned down.

3. Presentation: While the methodology and logical flow of arguments are very clearly presented, few original data are shown and some parts of the figures are difficult to resolve.

3.1 Original data: I am not sure whether a raw trace can illustrate phase coupling of 1/f noise or periodic oscillations. If this turns out impossible, at least 1/f plots with different slopes could be shown, maybe with multiple curves color-coded according to phase. In any case, the present data are mostly highly processed, hampering intuitive understanding.

3.2 Figures: First, several panels like Fig. 1b, Fig. 4b,c and others are very small (not a major issue in times of electronic publishing, but tedious for the reader). Second, the unit and scale of the radial axis of polar plots is often unclear to the (non-expert) reader, and

the insets in Figure 2 are very small and not sufficiently described. Third, the size of the differences between modulation of aperiodic and periodic activity, presented in Fig 4, is difficult to judge for most readers. Any way to highlight prominent cases (or, as a contrast, cases with small differences) or to make the deviation between 1/f and oscillation power more plastic/visible to the reader would be very welcome. Again, the parameters for this difference, as depicted in Fig. 4b (0-3) and 4c (-0.2 – 0.2) are not clear for most readers.

Specific comments (not ordered by priority):

4. The two locations (Münster, Leipzig) are frequently described as 'sites'. This is confusing to the naïve reader. Within the context of EEG/MEG and with the focus on the parieto-occipital region, many readers will associate 'site' or 'recording site' with a specific spot on the subject's head. I suggest using another word, e.g. referring to the two distinct teams or groups.

5. The content of Table 1 can be incorporated into the legend of Fig. 1.

6. Figure 1 and the Methods section indicate that power spectra are constructed from 1-40 Hz. The sliding window (time window 1 s, time steps 0.25 s) will probably not allow to measure reliably at low frequencies such as 1 Hz. Can you please comment and/or clarify?

7. The long discussion of alpha rhythms in the first paragraph of the introduction is distracting and may be partially moved to the discussion.

8. The discussion, on the other hand, could be shortened from ~4.3 to 2-3 pages at most.

9. 'IBB' and 'MPI' should not be used as figure labels (e.g., left and middle panels of Fig. 2). These are institutions, not biological entities!

10. Section 'Periodic and aperiodic components of...', second paragraph: were null beta weights really constructed with 500 tests or, rather, with 5000?

11. In the same paragraph (see comment #10) the use of 'slope and power' for aperiodic and periodic activity measures, respectively, should be made more explicit to avoid confusion.

Andreas Draguhn

Reviewer #3 (Remarks to the Author):

Summary:

In this paper, Kluger and colleagues study the relationship between respiratory cycles to fluctuations in cortical dynamics. Specifically, they assess whether the aperiodic exponent of the power spectrum of M/EEG recordings—as a surrogate of cortical excitation-inhibition balance— vary dynamically with the phase of respiratory cycles, using a novel time-frequency decomposition and parameterization tool (SPRINT). They find that, in both EEG and MEG data and across the cortex (and specifically in parietal-occipital regions), aperiodic exponent fluctuates with respiration peak and troughs. They repeat the analysis under several conditions, including per-electrode level, HCP parcellation-level, etc., and find that the effect is robust (along with appropriate statistical testing). Finally, they perform the same analysis but with aperiodic-removed oscillatory power, and show that aperiodic and oscillatory components have different respiration phase preference, suggesting different mechanisms of modulation.

Overall, I find the study to be well motivated given the gap in respiration-brain coupling literature, and also clearly written / presented. The application of SPRINT in this context is novel, and contributes interesting data on how respiration may be related to distinct neural processes in the aperiodic and oscillatory components of M/EEG. I have several concerns, however, regarding signal processing decisions, as well as interpretations and the lack of data on the modulation of aperiodic-removed alpha oscillation, especially in relation to the authors' previous work [ref 9]. Some other minor comments / suggestions are also included below.

- a note on nomenclature: I don't mean to be a prescriptivist, but I find the authors' usage of slope and exponent to be a bit confusing. In the specparam description of the M/EEG power spectrum, the aperiodic component follows a power law, i.e., power scales inversely with frequency, with a certain scaling exponent, $P \sim 1 / (f^\alpha)$. Alpha here would be the exponent of the aperiodic spectrum. This can be reformulated as a linear relationship in log-log, i.e. $\log P \sim -\alpha * f$, and alpha describes the slope of that line. Throughout the manuscript, the authors refer to this quantity, alpha, as the aperiodic or 1/f slope, and refer to changes as steeper or flatter slope, which is not problematic at all but perhaps warrants a short explanation upfront about the loglog transform. What is problematic is that the phrase "slope exponent" appears many times in the paper, which are basically two words that mean the same thing, but I take it to mean essentially "the magnitude of the aperiodic slope". I think everywhere it says "slope exponent" can just be replaced with "slope". For the rest of the review, I will follow the authors convention and use slope to mean the exponent of the power law aperiodic spectrum, where steeper slope refers to a larger exponent.

- another note, please include page number and line number, it's very difficult to refer to specific places in the manuscript otherwise (unless this is introduced in the submission process, if so please complain), and it will be equally difficult for the authors to find where I'm talking about in the manuscript.

1. clarifications and concerns regarding signal processing:

- if I understand correctly, the main pipeline is as follows: SPRINT outputs aperiodic and periodic parameter estimates from short time Fourier transforms with 250ms step length, resulting in a 4Hz slope time series (and oscillatory power estimates). The respiratory signal is recorded and peaks/troughs extracted, and the phase is linearly interpolated between peak to trough and trough to peak (I assume continuously?). Then, this respiration phase signal is "sampled" wherever there is a slope value, where respiration phase is first split into 60 bins, and the "effective respiratory phase" is a time series that is sampled at identical time points as the slope time series (so 4Hz), with value taken as one of the 60 possible bin values. Subsequent analyses average slope values that belong to the same phase bin, where permutation tests are also conducted. If this is correct, I have several concerns and questions:

- where it says "we collected all SPRINT outputs computed at respiration angle of $\omega \pm \pi/10$ ", I'm not really sure what this means? Does it mean the bin center is at ω , but the bin is actually $\pi/5$ wide, such that the 60 phase bins are overlapping?

- Similarly, "we computed individual bin-wise averages..., yielding quasi-continuous 'phase courses'...", and later, "we finally binned all SPRINT outputs into $n=60$... and computed the bin-wise average slope fit to yield a null time series...". Bin-wise averaging means to average all slope values that fall within that bin, i.e., all slope values that share a corresponding phase bin "ID"? If so, doesn't this mean it's collapsed across time already, so how is a "phase course" and "null time series" arrived at? My naive expectation would be

that there is simply two time series, both at 4Hz and at matching time points, where the slope is just the SPRINT output for that STFT window, and respiratory phase is the linear interpolation queried at that time point? Please clarify on these two points above.

- related, I really like Fig 1 that shows a schematic of the analysis pipeline. I think it could be a little clear. For one, the respiration signal has an arrow into the SPRINT box, but it doesn't really influence the M/EEG analysis in anyway if I understand the algorithm correctly. If anything, where there is slope estimates determines where the respiratory phase is sampled, such that the two form two parallel time series sample at the same time points. I believe Fig 1b shows this? Maybe one way to make it clearer is to show draw an arrow from the top part of a (the resp signal) into the "extract respiratory phase" part in b, and a parallel arrow from SPRINT to the top part of b (with some rearrangement). This is just a suggestion on how to potentially make it clearer to avoid confusions such as my two points above.

- I'm not familiar with the respiration DSP literature, but is it standard practice to find the peak and trough and simply linearly interpolate between the two for phase estimates? Why not just use the analytic phase (via Hilbert transform) since it's a relatively narrowband signal? The latter is what one would usually define as "phase", and compared to that, the former method could introduce substantial errors in phase estimate, and therefore binning, which could significantly affect the result of the study. Specifically, for a non-sinusoidal signal, the distribution of phases is not uniform around the circle if divided into equal bins. Alternatively, one could bin such that each of the 60 bins have the same number of samples, and would be equally justifiable. In fact, the authors should show the distribution of phases, i.e. a histogram with the 60 defined bins, and, importantly, in general discuss why linearly interpolating phase would be preferable to using an actual estimate from Hilbert transform. I would also recommend repeat at least a subset of the analyses using the Hilbert phase to assess whether such a choice would impact the result significantly.

- in general, I'm a bit confused at how the shuffling was done for the permutation tests. When it says "we shuffled respiration time course", does it mean shuffling the raw signal? the interpolated phase value? or something else? My naive guess is that the 4Hz signal (either the resp phase or slope) is shuffled across its time index, in which case this might be described more clearly? Furthermore, since naively shuffling destroys temporal correlation, the authors might consider using a method that creates a null time series while preserving autocorrelation (e.g., circular shifting, IAAFT, etc.)

- there were a few analyses that involved averaging across channels/sensors/sources before submitting the signal to SPRINT. In general, and especially when oscillatory signals are concerned, I would heavily recommend against averaging the raw signal across space, as it may result in destructive interference of a shared oscillation, resulting in a artificially lower estimate of power, especially if the oscillation itself is a traveling wave (as has been shown in many papers now). Similarly, it's not known how averaging would affect the broadband spectrum, but in general one can expect it to affect one part of the spectrum preferentially, e.g., averaging over white noise channels reduces overall power/variance, which could lead to decreasing power at higher frequencies (near the noise floor), hence affecting the slope estimate out of any specparam methods. Since in the per-electrode analysis, all the relevant slope and oscillation estimates were provided from SPRINT anyway, I would definitely recommend averaging at the slope estimate level, which could even result in less variance in the ROI slope estimate.

- it would be great to see some examples of the power spectrum in loglog, to see if a linear fit over the whole frequency range is appropriate. In many cases, we're seeing that the exponent estimate is biased by the shifting plateau (or knee) of the spectrum, even without

real spectrum changes (e.g., Gao et al 2020 elife). I know this sounds like I'm passive aggressively suggesting for the authors to cite our new papers that design evermore spectrum features (it's really not), but in all honesty this is just a suggestion to ensure that the correct interpretation is made wrt E-I balance, and it would also be interesting if timescale is also affected by breathing. On that note, the authors should cite Trakoshis et al 2020, elife, which has a much better computational model linking E-I changes to spectral slope.

2. clarifications on results:

- given that the authors' previous publication investigated alpha power and respiratory phase (presumably without specparam or SPRINT), I am very curious whether aperiodic-removed alpha power modulation is still associated with specific phases of respiration. Related, I don't really understand the justification to sum over the entire periodic spectrum, and while the authors did the analysis on per-band power, nothing is mentioned on that front other than that it's in Supp Fig 1. How come?

- in Figure 2, it looks like the significant regions of the radial histogram is slightly rotated between EEG and MEG, especially for the slope decrease regions. In addition, it looks like for the significant increase and decrease are (roughly) π degrees apart in the MEG dataset, but not in the EEG. Are there explanations or discussion for this?

- Comparing Figure 3a to figure 2, it looks like the histogram of mean phases (Fig3a) are rotated vs. the maximum bin-average slope magnitude (Fig2 red): is this meaningful, or due to the two different quantities computed? In general, do the circular distributions across these analyses agree with each other (statistically not significantly different)?

- Figure 3b shows many spots, especially in the EEG dataset, where the mean phase is near 0 instead of π . Is there an explanation for this?

- Figure 4b shows a distribution of differences of mean phase, but is not weighted by the actual means nor the significance of the means, correct? In other words, many channels could have very small mean phases (i.e. near uniform circular distributions) for both slope and power, but nevertheless result in a difference, is this be meaningful? What would happen if one only took histogram over parcels that were significantly non-uniform for either or both slope and oscillation? Apologies if I had missed where this was done.

- throughout the manuscript, it seems that the authors imply that breathing causally modulates slope (hence E-I balance), whereas the data only shows a correlation. This could be a misinterpretation on my part, but given the discussion on predictive processing and the volitional control of breathing, it seems that the authors have an implicit stance that breathing physically modulates cortical dynamics. I don't believe the current data supports this, as opposed to, say, a task where participants are asked to breathe at controlled time points. I believe this warrants some qualification in the discussion section.

Richard Gao, PhD
University of Tuebingen

Main changes

- updated and amended all 4 main figures to include reviewers' suggestions
- added a new main figure and 6 supplementary figures
- added new control analyses for
 - potential motion-related artefacts
 - influence of breathing rate and breathing depth
 - temporal autocorrelation in surrogate respiratory data
- re-analysed a previously published MEG data set (N = 28) to differentiate between normal and deep breathing
- conducted a new within-subject MEG study (N = 25) to differentiate between nasal and oral breathing

Point-by-point reply to reviewers

The reviewers' comments are printed in black. Our responses are printed in blue, red text indicates changes in the revised manuscript.

Reviewer #1

The manuscript by Kluger et al. uses recent new methodology for parametrizing the aperiodic and periodic components of the spectrogram of human neurophysiological time series, to study whether these latter are related to the phase of the respiratory cycle. This is a significant question because the latter is related to the sensitivity of sensory perception, while some characteristics of the aperiodic component of the neurophysiological frequency spectrum have been proposed to be related to neural excitability. The temporal alignment between independent observations derived from human brain and body physiology opens meaningful new perspectives on perception and related behaviour, and would inform future studies of their trial-to-trial fluctuations.

The manuscript provides sufficient background motivation to the reported study. One strength is to report data pooled from two different sites, each providing distinct modalities (MEG vs. EEG), with a reasonable N of about 40 participants each.

The manuscript is overall well written, but there are several instances where the approach is unclearly reported, both in the text and the figures, and where the methodology needs to be revised or at least, better justified.

Taken together, I believe the manuscript has potential but also shortcomings. It may better fit a more specialized journal.

Major points:

- Several of the null models used for testing the linear mixed effect models resort to randomized individual respiration time series. It is crucial that the randomized versions of the respiration phase preserve the original signal's temporal autocorrelation, to avoid deflating the specificity of the resulting tests.

We thank the reviewer for raising the issue of temporal autocorrelations in our surrogate distributions. To address this comment (and following a related suggestion by Reviewer 3, see below), we have now recomputed the null distributions of 1/f slope in Fig. 2 by using an iterated amplitude-adjusted Fourier transform (IAAFT; Theiler et al., 1992) of the respiratory trace instead of shuffled respiration. This procedure iteratively generates a surrogate time series (in

our case, surrogate respiratory data) with a temporal autocorrelation closely matched to the original signal. While there are slight changes in the timing of the modulation effects (see Fig. 2) compared to the initial submission, the overall pattern of results remains unchanged. The corresponding figure and paragraphs have been revised accordingly.

Critically, all remaining analyses (particularly, all LMEMs) were not based on shuffled respiration time series, but on shuffled courses of slope/ power ~ phase bins, which destroys the systematic relationship between e.g. slope and respiration phase.

We do acknowledge that our description of the null distribution computations were not sufficiently clear and we apologise for the confusion this may have caused. The description in the Methods section has been revised and now reads as follows:

To test whether $1/f$ was significantly modulated over the ROI sensors, we computed $k = 5000$ random iterations of the LMEM by shuffling bin-wise $1/f$ slope fits on the individual level. This way, any meaningful relations between $1/f$ slope and respiratory phase were removed, thereby specifically testing the hypothesis that $1/f$ slope changes significantly with respiratory phase. For each of these iterations, we once again computed the phase vector norm from the resulting beta weights, yielding a random distribution of $k = 5000$ 'null vector norms'. Significance of the empirical vector norm was determined by its percentile relative to this null distribution.

As shown in Figs. 2 and 3, for the sensor-level analyses, SPRiNT parameterisation was conducted on individual MEG sensors (Münster) and EEG channels (Leipzig). For the analysis of $1/f$ slope modulations within the parieto-occipital ROI, we computed mean SPRiNT estimates across the corresponding sensors or channels, respectively. LMEMs were then computed on ROI-average estimates for $1/f$ slope ~ respiration phase for each lab individually (Münster, Leipzig) as well as for the pooled data from both labs (see Fig. 2). For the whole-scalp analyses shown in Fig. 3, LMEMs were computed on single-sensor/channel SPRiNT estimates.

Having established an overall significant influence of respiratory phase on $1/f$ slope with the LMEM approach described above, we next aimed to characterise *when* (i.e., at which respiratory phases) $1/f$ slope was significantly steeper or flatter. To this end, we implemented a permutation approach as follows: For each participant, we constructed a surrogate respiration time series using the iterated amplitude-adjusted Fourier transform (IAAFT⁵⁵). In contrast to shuffling the respiration time series, this iterative procedure preserves the temporal autocorrelation of the signal, which is critical for constructing a fitting null distribution for permutation testing. From these IAAFT-transformed respiration time series, we extracted the surrogate respiratory phase values corresponding to each slope estimation from SPRiNT. In keeping with the approach above, we finally binned all SPRiNT outputs into $n = 60$ equidistant, overlapping phase bins covering one entire respiratory cycle ($-\pi$ to π) and computed the bin-wise average slope fit to yield a 'null time series' of $1/f$ slope over surrogate respiration phase. For each participant, this procedure was repeated 5000 times and resulted in a null distribution of 5000 surrogate $1/f$ slope estimates x 60 phase bins. (pp. 25-26)

- MEG head movements have been rightfully corrected for and regressed out of the analyses. Yet, to ascertain that artefactual residuals did not contaminate the analyses, at least one control test should be provided in Supplementary Material substituting the respiration time series with the principal component (from PCA) of all recorded movement time series.

We appreciate the reviewer's comment regarding a clear demonstration that our results were not influenced by motion artefacts. The analyses they suggest, however, is unfortunately not well-suited to show that motion did not play a role: All signals extracted by a PCA on movement time series were in fact removed prior to any analyses and can't possibly be a confound - this analysis would merely show *how movement would have influenced the results had they not been removed*. As they were in fact removed entirely, there are no 'movement-related residuals' in the MEG data to apply such an analysis to.

As we were eager to address the reviewer's comment, we conducted a new analysis to characterise the relationship between even minute muscle artefacts and respiration. Specifically, for the ROI analysis shown in Fig. 2, we high-pass filtered our data using a 4th-order (forward and reverse) Butterworth filter with a cut-off frequency of 60Hz to amplify muscle-related contaminations. We then subjected the filtered data to ICA analysis (extracting the first 20 components). Each extracted component was individually scanned for the occurrence of muscle activity using the automatised Fieldtrip function *ft_artifact_muscle* (see http://new.fieldtriptoolbox.org/reference/ft_artifact_muscle/ for documentation). This function applies an 8th-order Butterworth bandpass filter (110 - 140Hz) followed by computation of smoothed amplitude envelope using the Hilbert transform. Muscle artefacts are then identified as events exceeding a default threshold of $z = 4$ in normalised data. Using this function, we determined the number of muscle artefact events within each ICA component. We defined the component with the highest number of events as the 'artefact component' (containing an average of 3.2 ± 1.3 artefact events [$M \pm SD$]). On the group level, the mean (Fisher z-transformed) correlation between individual 'artefact components' and respiration time series was virtually zero ($M_{rho} = .0063$). For such 'artefacts' to have any meaningful influence on our MEG results, they would have to be substantially coupled to the respiration signal, which was obviously not the case due to the thorough removal of movement-related signal by our regression approach. Another clear indication that movement artefacts did not influence our MEG results is how consistent they are with the EEG results obtained in Leipzig. As EEG does not suffer from the potential interference of head movements, they are well-suited as a control case for our MEG data: Had MEG-specific movement artefacts influenced our effects, they would be present in MEG, but not in EEG. The highly consistent pattern of results between MEG and EEG (as evident from Fig. 2 and the non-parametric two-sample test for circular data (see below) further increases our confidence that our complex regression approach successfully removed movement-related signal from our MEG data.

- The conclusions (p.6) that respiratory phase influence the aperiodic neural signals is incorrect. The linear relationship of the mixed effect model does not establish a causal influence, simply an association.

We appreciate the reviewer's suggestion to phrase some of our main conclusions more cautiously. In line with their remarks, we have rephrased several paragraphs throughout the manuscript to more adequately reflect the associative nature of our results.

For both MEG and EEG as well as for the pooled data, the empirical group-level beta weight for respiratory phase was significantly higher than the respective null beta weights (all $p < .001$), strongly indicating a significant overall association between respiratory phase and aperiodic neural signals over posterior cortices (Fig. 2). (p. 5)

Of the 23 parcels of interest, 18 parcels showed a significant association between respiration and 1/f slope. (p. 11)

In case the respective LMEM did not indicate significant association with either slope or power within any given parcel, the corresponding time series are drawn with reduced opacity. For significantly associated parcels only, red and blue triangles indicate respective maxima and minima over the respiratory cycle. (p. 12)

In the present data, the preferred phase differences between source-level periodic and aperiodic modulations within the RMBO network show at least partly differential dynamics between both components, suggesting a more complex coupling of neural signalling to the respiratory rhythm than previously assumed. (p. 15)

If 1/f slope and oscillatory power were similarly coupled to the respiratory rhythm, their respective courses over respiratory phase would not differ and the phase difference between their circular means would be (close to) zero. In contrast, systematically different distributions of slope and power would suggest differential dynamics in their respective coupling to respiratory phase. (p. 25)

- The rationale for presenting the pooled data in addition to the two datasets analyzed separately needs to be clarified. What do we learn from pooling the two datasets?

Our motivation for pooling the MEG and EEG data sets was twofold: First, we wanted to highlight the commonalities between both modalities, i.e. characterise the results jointly found in both data sets. Second, pooling essentially doubled the number of participants, thus substantially increasing the statistical power of the overall analysis.

In order to clarify our motivation, we have amended the revised manuscript as follows:

Subsequently, we pooled both data sets in order to capitalise on higher statistical power and to illuminate phase-locked modulations jointly found across modalities. (p. 5)

- Some key aspects of Figure 2 are unclear. The “concentric columns” referred in the main text look like concentric dashed lines that are virtually identical across all phase bins. Further, the black lines supposed to show the 5th and 95th percentiles are not seen in the figure. Further, it is unclear what the inner dark circle represents in each polar plot. Also, contrarily to the text statements, the blue markings are in the early phase of expiration for EEG, not the late phase of inspiration.

We cordially thank the reviewer for giving us the opportunity to clarify central aspects of Fig. 2. The ‘dashed lines’ they refer to are in fact concentric scatter plots in which each dot represents a phase-specific mean of the respective null iteration (apologies for the potentially confusing term ‘columns’ which has now been replaced). Since null iterations are computed on IAAFT transforms of the original respiratory trace, similarities across phase bins are to be expected (although slight differences in the null distributions are noticeable upon close inspection). Finally, the ‘dark circles’ mentioned by the reviewer are precisely the 5th and 95th percentiles of the null distributions.

We apologise for the lack of clarity in this respect and have amended the figure and its legend to more clearly illustrate the confidence intervals of the surrogate distributions:

Fig R1. Group-level respiration-locked modulation of aperiodic 1/f slope over parieto-occipital sensors. Respiration phase was extracted for a total of $n = 60$ overlapping phase bins covering the entire respiration cycle between subsequent inhalations. Overall significance of respiration phase-locked modulations was assessed by means of permutation statistics: For each data set, the combined LMEM beta weight for the phase vector norm (i.e., $\sqrt{\sin^2 + \cos^2}$) is shown in light blue against a distribution of null betas ($k = 5000$, top right corners). Within each polar plot, coloured dots show respiration phase-dependent 1/f slope gained from SPRiNT computations on periodic spectra over parieto-occipital MEG sensors or EEG channels, respectively. Concentric scatter plots (black dots) indicate null distributions of group-level mean exponent values computed for each phase bin ($k = 5000$ iterations). Solid black lines indicate the 5th and 95th percentile of each bin's null distribution. 1/f slope was consistently flatter around the inspiration-to-expiration transition (blue markings) and steeper around the expiration-to-inspiration transition (red markings). In = inspiration, ex = expiration. Shown as panel a in Fig. 2 of the revised manuscript.

The reporting of significant slope decreases has been corrected:

For the pooled data from both labs, we observed a steeper slope (indicating stronger inhibitory activity) around the expiration-to-inspiration transition (ranging from -156° to 143° and around 131°). Conversely, the slope decreased (i.e., flatter slope indicating stronger excitatory currents) during the inspiratory phase (between -88° and -64°) and shortly after the inspiration-to-expiration transition (9° to 46° , see Fig. 2). (pp. 5-6)

The slope of aperiodic neural activity was not uniform over the respiratory cycle, but systematically modulated in such a way that 1/f slope was flattened around the inspiration-to-expiration transition and steeper around the expiration-to-inspiration transition. (p. 9)

- 'exp' is an ambiguous contraction as it could be interpreted as 'exponent' in the present context (this is a minor point).

We thank the reviewer for making us aware of this potential ambiguity. In line with their comment, all instances of 'insp' / 'exp' have been substituted with 'in' / 'ex' throughout the manuscript.

- How to read Table 1 is entirely unclear, despite the caption. The layout needs to be clarified. As a more general comment, testing the phase disparity between the MEG and EEG datasets needs to be conducted in a quantitative fashion and discussed.

Following a suggestion by Reviewer 2, Table 1 has been removed and the information therein can now be found in the legend of Fig. 2. In order to address the reviewer's second comment, as suggested, we have now added a statistical comparison of MEG and EEG slopes. A non-parametric two-sample test for circular data (akin to a Kruskal-Wallis test for linear data) revealed that the circular means of MEG and EEG slope ~ respiration courses were not significantly different from one another ($P(77) = 0.21$, $p = .651$). The Results section of the manuscript has been amended to include this information as follows:

Consistency across MEG and EEG data was supported by a non-parametric two-sample test for circular data (akin to a Kruskal-Wallis test for linear data), which showed that the circular means of MEG and EEG slope ~ respiration courses were not significantly different from one another ($P(77) = 0.21$, $p = .651$). (p. 6)

- P.8: This notion of shift is not well explained in the text main body. I suppose this represents a form of bias indicating around which respiratory phase the slope tends to be the strongest? The rationale and definition for this measure need to be entirely clear from the text.

We thank the reviewer for pointing out our insufficient explanation of the term 'shift'. Exactly as they speculate, we meant to describe a certain respiratory phase at which 1/f slope would be systematically steeper or flatter. The reviewer's comment has made clear that 'shift' is not the ideal description, which is why we have now replaced it with the term 'modulation' throughout the manuscript.

- Figure 3: The topographies look noisy and not evidently consistent between the MEG and EEG data, which questions the significance of the data interpretation. Also, the strength of the slope course is a notion that is not clearly explained. The colorscale of Panel a shows a sens/chan unit, which meaning is also unclear.

Thank you for bringing our attention to a few ambiguities in Figure 3. First, as single sensor/channel slope courses are shown, the colour scale in panel a does indeed show the range of MEG sensors (1-275) and EEG channels (1-64). We apologise for the insufficient previous description and have revised Figure 3 and its caption accordingly. Moreover, 'strength of slope modulations' was meant as a measure of effect size: As stated in the figure caption, the empirical LMEM weights are depicted in units of standard deviation of their respective null distribution. This allows the reader to see which sensors/channels showed the biggest fluctuations (i.e., the strongest modulations) of 1/f slope over the respiratory cycle. For the sake of clarity, the term 'strength of slope modulations' has been replaced with 'effect size' throughout the manuscript and the figure caption now features an extended explanation.

Fig. 3. Single-sensor (MEG) and channel (EEG) modulation of 1/f slope. **a**, Polar plots shows group-level averages of normalised single-sensor/channel slope courses over the respiration cycle. LMEM revealed 1/f slope over respiration to significantly deviate from a uniform distribution in $n = 240$ out of 275 MEG sensors and $n = 53$ out of 64 EEG channels. Only significant MEG sensors (top) and EEG channels (bottom) are shown, colour coding illustrates sensors and channels 1 to n . Polar histograms illustrate circular mean directions of single-sensor/channel slope distributions (colour-coded according to respiration phase). **b**, Topographic representations of significant modulations and their phase direction for MEG sensors (top) and EEG channels (bottom). **c**, Population statistics quantifying the effect size of slope modulations seen in **b**. For each sensor/channel, we show the vector norm of LMEM beta weights for sine and cosine of the respiratory phase (akin to a harmonic regression approach). These vector norms were normalised using the mean and standard deviation of the null distribution and are thus given in z values (see Methods for details). In = inspiration, ex = expiration.

Finally, regarding the consistency between MEG and EEG results, we would like to briefly elaborate on the comparison of both data sets and the purpose of Fig. 3 itself: Following the ROI-based analysis shown in Fig. 2, we extended the scope towards a full-scalp analysis in order to illuminate whether the reported slope modulation pattern would generalise to other channels, how strong these modulations are across the scalp, and whether these results would be similar in both data sets. When evaluating these consistencies across MEG and EEG data, it is worth noting that, due to the rotated sensitivities of MEG and EEG for source orientation, some cross-methods differences between any two topographies are very much to be expected. We would like to stress that, in line with the crossmodal posterior effects reported in Fig. 2, both the clustering of circular means around $\pm \pi$ (Fig. 3a) as well as the parieto-occipital focus of highest effect sizes (Fig. 3c) are highly consistent between MEG and EEG topographies.

- Figure 4: Overall, the figure is very busy and hard to read. More importantly, the similarity of the shape of the slope and power polar plots is truly not convincing enough. This is emphasized by the only scattered occurrences of when such similarities are statistically significant along the unit circle.

We are grateful for the reviewer's remarks and would like to use the opportunity to clarify several aspects of Fig. 4. First, one central message of the figure is precisely the *lack of similarity* between slope and power plots within each parcel. In fact, panel b shows a dedicated

analysis of the group-level phase differences between both measures, with additional statistics provided in the main text. Consequently, there is to be no indication of ‘statistical significant similarities’ (or differences) in the figure. Most likely, the red and blue triangles in the polar plots have caused some confusion in this regard - as described in the figure caption, they indicate the respective maxima and minima of power and slope over the respiratory cycle. For the sake of clarity, the figure has been amended to include a legend disambiguating these symbols (see panel a, bottom left):

Fig. 4. Source-level ROI modulation of 1/f slope and accumulated oscillatory power in the MEG. a, For each cortical node of the RMBO network², we show the group-level average courses of 1/f slope (left polar plots) and accumulated spectral power after removal of the aperiodic component (1 - 40 Hz; right polar plots) over the respiratory cycle. For ROIs comprising more than one parcel from the HCP atlas, slope and power courses are superimposed for each individual parcel. In case the respective LMEM did not indicate significant association with either slope or power within any given parcel, the corresponding time series are drawn with reduced opacity. For significantly associated parcels only, red and blue triangles indicate respective maxima and minima over the respiratory cycle (see legend). **b,** Polar histogram shows phase differences between the circular means of slope and power courses across parcels. The clear majority of non-zero phase differences were corroborated by Watson's U² statistics indicating distinct temporal dynamics of slope and power modulations within each parcel (see main text). **c,** Centre: For each ROI, we show violin plots of single-participant U² statistics (slope/power vs uniform distribution, respectively). Higher U² statistics indicate stronger non-uniformity. Left: Exemplory slope modulations of SMA and OFC (normalised across respiration phase for illustration purposes). SMA was more strongly non-uniform overall (see centre), which is visible from both stronger maxima (Δ_{max}) and minima (Δ_{min}) compared to OFC. Right: Exemplory power modulations of OFC and I. INS. For both ROIs, we superimpose power

spectra (with $1/f$ removed) from all 60 phase bins. Overall, power varied more strongly with respiratory phase within OFC compared to I.INS (see centre). TPJ = temporo-parietal junction, ITG/STG = inferior/superior temporal gyrus, ANG = angular gyrus, FEF = frontal eye-field, TP = temporal pole, SMA = supplementary motor area, OFC = orbitofrontal cortex, CUN = cuneus, LS = lingual sulcus, ACC/PCC = anterior/posterior cingulate cortex, INS = insular cortex. In = inspiration, ex = expiration.

Minor points:

- Figure 1 is a cartoon illustration of the approach, yet it is confusing. The illustration of the “aperiodic slope” in the center frame does not show a log-log plot of the aperiodic spectrum, and therefore not a straight line representation of the latter. Hence the notion of slope is meaningless here.

Symmetrically, the periodic spectrum shows the full parametrized spectrum, which includes the aperiodic component.

Therefore Figure 1 requires changes to make sure the figure didactically and accurately informs the reader about the approach.

I believe the peer-reviewed version of SPRiNT just published in eLife provides a spectrogram representation of the periodic and aperiodic parameter estimates. This may be considered to clarify their time-varying nature in this figure and maybe in the main body of results of the paper.

We thank the reviewer for their remarks to improve the clarity of our methods figure. Following their suggestions (and those of Reviewer 3, see below), we have now revised several aspects of Fig. 1:

Fig. 1. Synopsis of acquired data and applied methods. **a**, In two labs, we simultaneously acquired nasal respiration as well as eyes-open, resting-state MEG (IBB, Münster) and EEG data (MPI, Leipzig) in continuous 5-min recordings. After preprocessing, single-sensor/-channel M/EEG data (middle panel) were subjected to the SPRiNT algorithm²⁶ (top). Here, using a moving-window approach (window length = 1 s, 75% overlap between neighbouring windows), spectral components of neural time series are estimated using a Fast Fourier transform. These frequency-domain data are then parameterized using the *specparam* algorithm²⁵ which yields both aperiodic and periodic components of neural activity in that time window (see inset). Repeating this procedure along the entire recording thus yields time-resolved fits of the aperiodic $1/f$ slope as well as time-resolved periodic spectra ranging from 1 - 40 Hz. Respiratory phase was computed using two-way interpolation (int) of the normalised raw respiration signal (peak-to-trough, trough-to-peak; bottom). **b**, For each time point used as a moving-window centre in the SPRiNT algorithm, we then extracted the corresponding respiratory phase. This allowed us to sort all time-resolved slope fits and periodic spectra according to the respiratory phase at which they were computed. In keeping

with previous work ^{2,9} we finally partitioned the respiration cycle into $n = 60$ equidistant, overlapping phase bins and computed bin-wise averages of slope fits and periodic spectra. This approach thus yielded quasi-continuous, respiration phase-resolved courses of both periodic and aperiodic components of brain activity for each sensor/channel within each participant.

As suggested, the 'output' section is now similar to the original publication by Wilson et al. (eLife, 2022) and includes a clearer depiction of the SPRiNT output. One exemplary 'slice' of this output is shown in detail to illustrate the original power spectrum as well as the full SPRiNT model fit and the aperiodic fit. Finally, the illustrations of the time-resolved slope fits and spectra have been changed according to the reviewer's suggestions.

Reviewer #2

Kluger and coworkers have analyzed the effects of respiration on rhythmic and non-rhythmic activity in the human neocortex. Using MEG- and EEG-recordings, respectively, from two cooperating groups, the authors show that the slope of the spectrogram ($\log(\text{power})$ vs. $\log(\text{frequency})$) varies along the breathing cycle in resting humans. Moreover, they find a differential respiratory modulation of the non-rhythmic versus the rhythmic components of the power spectrum. Based on recent work on aperiodic brain activity (esp. from B. Voytek and colleagues) they interpret their findings as evidence for respiration-dependent fluctuations in excitation-inhibition cycle. Furthermore, they interpret the different cycle-dependent modulation of oscillatory (periodic) versus non-oscillatory (aperiodic) activity as evidence for different underlying mechanisms.

The groups have long track records in human electrophysiology, and the paper adds to previous work along the same line (Kluger et al., eLife 2021). The question is certainly relevant and I share the author's impression that it has not been studied before - analyzing E/I-balance by using the $1/f$ -slope of spectrograms is a relatively new approach (Gao et al., 2017), and connecting this measure with the breathing cycle requires using new and advanced non-stationary analysis tools (Wilson et al., BioRxiv 2022). Thus, the present contribution makes a significant contribution to a new line of research focusing on non-oscillatory brain activity, a highly understudied field in network- and cognitive neurosciences.

Despite this very positive general impression, I have some conceptual and editorial issues:

1. Lack of interventional or correlative manipulations. At present, the study appears a bit like a highly skillful application of the new analysis routine, while the biological significance of the findings does not become entirely clear. The electrophysiological observation seems to be robust: most brain regions are modulated by respiration; this applies to both for $1/f$ and cumulative oscillatory power; the phase-wise modulation differs between both constituents of electrographic activity. However, we learn little about the behavioral or cognitive relevance, state-dependence, modulation by altered breathing etc. Of course, the authors cannot test all these factors together with the first description. However, if the phenomena were sensitive to any intervention, confidence in the biological/cognitive function would increase and hypotheses regarding the mechanism would be substantiated. For example: is there any correlation with breathing parameters (frequency, depth, mouth versus nose)? Is the phenomenon altered (at least in pilot experiments) when subjects leave the default state and concentrate on a task, e.g. visual? Is there any correlation with arousal? These questions are just examples – the critical point is that any such experiment could bring the data from a 'mere'

description (not meant as a pejorative term) to a level stimulating further studies. Note: the feeling that ‘something is lacking’ may partly result from the somehow unimpressive presentation of data (see my comment #3).

We appreciate the reviewer’s well-justified critique which we took as an opportunity to expand our analytical scope towards an exploration of potential biological underpinnings. The reviewer particularly touches on two related, yet independent aspects with regard to respiratory mechanics: On the one hand, parameters like frequency and breathing depth characterise individual respiration patterns irrespective of state changes or respiratory pathways. The question of nasal vs oral breathing, on the other hand, gives rise to a central distinction between different anatomical pathways (e.g., involving OB stimulation for nasal, but not for oral breathing). In what follows, we would briefly like to elaborate on both aspects separately and introduce the corresponding novel analyses now included within the revised manuscript. First, for both data sets of the present study, we investigated the correlation between 1/f slopes and breathing rate and depth, respectively. Breathing frequencies were extracted as average peak-to-peak distances during the peak detection procedure. Breathing depth was extracted as the integral of the respiration time course, normalised by the number of breathing cycles (to remove breathing frequency as a potential confound). For the posterior ROI analysis shown in Fig. 2 of the manuscript, this analysis yielded one group-level correlation coefficient for ROI-average slope and breathing frequency/depth per data set. Neither breathing frequency nor breathing depth were significantly correlated with mean slope in either data set (all $p > .079$). Fig. R2 shows these correlations for MEG and EEG data:

Fig. R2. Correlation between mean 1/f slope and respiratory parameters. For both MEG (left) and EEG data (right), we plot group-level correlations of 1/f slope and breathing rate (top) as well as breathing depth (bottom). Neither respiratory parameter was significantly modulated with overall steepness of the aperiodic component. Shown as panel c in Fig. 2 of the revised manuscript.

This control analysis is described in detail in the revised Methods section of the manuscript:

In order to assess the potential influence of individual breathing parameters on aperiodic fluctuations, we computed breathing rates and depths for each participant. Breathing rates were extracted as the mean distance between inspiratory peaks as defined by the peak detection algorithm described above. Breathing depths were computed as the integral of the individual respiration time series, normalised by the number of breathing cycles during the recording. (p. 23)

While these analyses were certainly instructive, their interpretability is somewhat limited by the small amount of intraindividual variation in the respiratory time series - within each subject, natural breathing is simply too consistent to conclusively investigate whether different breathing 'modes' (e.g., deep vs normal breathing) would influence respiration-locked modulations in $1/f$ slope. Therefore, we next re-analysed a previously published data set ($N = 28$) in which we had recorded 5 minutes of whole-head resting-state MEG during both deep and normal nasal breathing (for details, see Kluger & Gross, *NeuroImage* 2020). Recording, preprocessing, and analysis of these data was otherwise identical to the ROI analyses described in the revised manuscript. Demographic and methodological details can be found in the revised Methods section:

Participants and data acquisition (control MEG study for deep vs normal breathing). The MEG sample for the comparison of deep vs normal breathing has previously been published elsewhere³. Twenty-eight volunteers (14 female, age 24.8 ± 2.87 years [mean \pm SD]) participated in the study conducted at the Institute for Biomagnetism and Biosignal Analysis in Münster. All participants reported having no respiratory or neurological disease and gave written informed consent prior to all experimental procedures. The study was approved by the local ethics committee of the University of Münster (approval ID 2018-068-f-S). MEG recording parameters and procedures were identical to the original recordings described above.

Data were acquired in two 5-minute runs with a short intermediate break (determined by the participants). Within each run, participants were instructed to either breathe normally or voluntarily deeply (with their mouths closed) while maintaining their normal respiration frequency, i.e. not to breathe more slowly during blocks of deep breathing. Continuous monitoring via video ensured participants did indeed keep their mouth closed throughout the recording. The order of deep and normal breathing was counterbalanced across participants. For both runs, we recorded the respiratory signal as thoracic circumference by means of a respiration belt transducer (BIOPAC Systems, Goleta, USA) placed around the participant's chest. Individual respiration time courses were visually inspected for irregular breathing patterns such as breath holds or unusual breathing frequencies, but no such artefacts were detected. (p. 22)

The comparison of deep vs normal nasal breathing revealed that $1/f$ slope over the posterior ROI was indeed more strongly modulated during deep (compared to nasal) breathing ($z(27) = 3.02, p = .003$):

Fig. R3. Comparison of parieto-occipital 1/f slope ~ respiration phase during normal and deep nasal breathing. Data from the two breathing conditions were collected in a previously published MEG study (Kluger & Gross, 2020; see Methods for details). Left panel shows exemplary respiration time series of both conditions for a single participant. Centre panel shows individual median depths across conditions, confirming that participants did in fact breathe deeper during the deep breathing condition ($z(27) = 6.19, p < .001$). Individual ranges of slope ~ respiration (as a measure of modulation strength) were significantly greater during deep (vs normal) breathing ($z(27) = 3.02, p = .003$; right panel), indicating a stronger modulatory effect of deep breathing on posterior 1/f slope. The figure is shown as panels d and e in Fig. 2 of the revised manuscript.

Third, we compared respiration-locked 1/f slope fluctuations at rest during nasal vs oral breathing. To this end, we conducted a new study ($N = 25$) in which we simultaneously recorded six minutes of whole-head resting-state MEG and respiration (nasal, oral). During the oral breathing condition, participants wore a nose clip preventing them from breathing through their nose. Recording, preprocessing, and analysis of these data was otherwise identical to the analyses described in the revised manuscript. Demographic and methodological details can be found in the revised Methods section:

Participants and data acquisition (control MEG study for nasal vs oral breathing).

Twenty-five right-handed volunteers (10 female, age 26.3 ± 3.3 y (mean \pm SD)) participated in the control MEG study conducted at the Institute for Biomagnetism and Biosignal Analysis in Münster. All participants reported having no respiratory or neurological disease and gave written informed consent prior to all experimental procedures. The study was approved by the local ethics committee of the University of Münster (approval ID 2021-785-f-S). MEG recording parameters and procedures were identical to the original recordings described above.

For each participant, we recorded two 6-minute runs of resting state activity: In one run, participants were instructed to breathe naturally through their nose. Continuous monitoring via video ensured participants did indeed keep their mouth closed throughout the recording. In the other run, participants were instructed to breathe through their mouth while wearing a nose clip to prevent nasal breathing. The order of nasal and oral breathing was counterbalanced across participants. Again, we recorded the respiratory signal as thoracic circumference by means of a respiration belt transducer (BIOPAC Systems, Goleta, USA) placed around the participant's chest. Individual respiration time courses were visually inspected for irregular breathing patterns such as breath holds or unusual breathing frequencies, but no such artefacts were detected. (p. 22)

For both respiratory conditions, we repeated the ROI analysis of parieto-occipital slope changes as described in the main manuscript. Within the new sample, the LMEM confirmed significant $1/f$ slope changes over respiration for both nasal and oral breathing (both $p < .001$). Replicating the effects from our MEG and EEG data sets, the new data again were overall characterised by steeper slopes during inspiration / around the expiration- to-inspiration transition and flattened slopes around the inspiration-to-expiration transition. A comparison of individual ranges of slope \sim respiration (i.e., $\max_{\text{slope}} - \min_{\text{slope}}$ over the respiratory cycle) revealed that the strength of respiration-locked $1/f$ slope modulations did not significantly differ between nasal and oral breathing ($z(24) = 0.80, p = .426$):

Fig. R4. Comparison of parieto-occipital $1/f$ slope \sim respiration phase during nasal and oral breathing. Data from the two breathing conditions were measured in a follow-up control MEG study (see Methods for details). Using the identical experimental setup and analyses as in the original MEG data, LMEMs confirmed a significant influence of respiratory phase on $1/f$ slope for both breathing conditions (see histograms). Group-level $1/f$ slope \sim respiration courses were similar to our previous MEG and EEG results (left) while the individual ranges of slope \sim respiration (as a measure of modulation strength) did not differ between conditions ($z(24) = 0.80, p = .426$, right). The figure is shown as panels f and g in Fig. 2 of the revised manuscript.

The Results section now includes a new subsection to report all new control analyses related to respiratory parameters:

Aperiodic modulations as a function of respiratory parameters. We conducted different control analyses to assess whether $1/f$ slope and its modulations covaried with parameters like breathing rate, depth, or route (i.e., nasal vs oral breathing). First, in the present data, individual $1/f$ slopes (averaged across the respiratory cycle) were not significantly correlated with breathing rates (MEG: $r(40) = -.17, p = .297$; EEG: $r(38) = .13, p = .418$) or breathing depths (MEG: $r(40) = .19, p = .230$; EEG: $r(38) = -.09, p = .591$) in either data set (Fig. 2c). However, the interpretability of these results is somewhat limited by the small amount of intraindividual variation in the respiratory time series - natural breathing is simply too consistent to conclusively investigate whether different breathing 'modes' (e.g., deep vs normal breathing) would influence respiration-locked modulations in $1/f$ slope. Therefore, we next re-analysed a previously published data set³ in which we had recorded 5 minutes of whole-head resting-state MEG during both deep and normal nasal breathing (see Fig 2d). In these data, a comparison of deep vs normal nasal breathing revealed that $1/f$ slope over the posterior ROI was indeed more strongly modulated during deep (compared to nasal) breathing ($z(27) = 3.02, p = .003$, see Fig. 2e).

Finally, we compared respiration-locked 1/f slope modulations at rest during nasal vs oral breathing in a new sample of $N = 25$ MEG data sets (see Methods). Repeating the posterior ROI analysis described above, respective LMEMs confirmed significant coupling of respiratory phase to 1/f slope during both nasal and oral breathing (both $p < .001$; Fig. 2f). The modulatory pattern over phase was again similar to the original findings (with certain limitations due to reduced power): During nasal breathing, 1/f slope was flattened during the inspiratory phase (-95° to -52°) as well as shortly after the inspiration-to-expiration transition (around 36°) and steeper around the expiration-to-inspiration transition (143° to -150°). During oral breathing, 1/f slope was flattened around the inspiration-to-expiration transition (-64° to 52°) and steeper around the expiration-to-inspiration transition (131° to -125°). A comparison of individual ranges of slope ~ respiration (i.e., $\max_{\text{slope}} - \min_{\text{slope}}$ over the respiratory cycle) revealed that the strength of respiration-locked 1/f slope modulations did not significantly differ between nasal and oral breathing ($z(24) = 0.80$, $p = .426$). (pp. 8-9).

Finally, we have revised the Discussion to highlight some of the reviewer's suggestions for subsequent studies:

The present findings open multiple avenues for future research, of which we want to briefly sketch three main directions. First, mechanistic advances could be made in a replicative study which includes a contrast of nasal vs oral respiration. The absence of phase-amplitude coupling driven by nasal airstreams during oral breathing could conceivably reveal to what extent aperiodic changes rely on and/or interact with oscillatory changes. While we speculate that respective changes in 1/f slope and oscillatory power could point to distinct modulatory pathways, targeted follow-up studies with sufficient statistical power are of critical importance to extend our initial results. Second, as outlined above, it would further be highly instructive to complement existing evidence of task-related 1/f slope changes^{23,54} by investigating their link to respiration in behavioural cognitive or perceptual paradigms. Timed breathing (in healthy participants or ventilated patients) or contexts involving different brain states, e.g. resting-state vs task, during attentional or arousal manipulation (including hyperventilation), or during sleep stages, could potentially unravel the link between breathing and fluctuations of (non-)oscillatory brain activity. (p. 16)

2. Interpretation. The observational character of the study prompts the authors to build mechanistic hypotheses, which is a perfectly valid approach. Two of these hypotheses seem problematic to this reviewer:

2.1 The respiration-dependence constitutes a somatic feedback signal. I agree that there is ample evidence for such somatic feedback (as quoted by the authors) but, at the same time, it is not clear that purely neurogenic mechanisms, starting from central pattern generators in the brainstem, do not play a role (see the quoted paper by Karalis and Sirota). This may even differ between brain regions or between aperiodic and periodic activity. Therefore, the feedback hypothesis should be put forward with some more caution, and the alternative mechanism(s) should be made more explicit. Ideally, changing from nasal to mouth respiration would be a simple and instructive experiment to add (see my comment #1).

As a result of this revision (particularly the addition of new data on nasal vs oral breathing), we have rephrased the feedback hypothesis in the Discussion section towards a more cautious interpretation:

Hence, there are (at least) two different ways in which neural activity could be coupled to the act of breathing. It may be the case that the differential coupling of respiration to periodic and aperiodic activity reflects these distinct mechanisms. However, the evidence so far (including presented data from the control study) neither precludes the possibility that both types of modulation could be rooted in the same underlying mechanism. Therefore, critical future work is needed to further elucidate the functional and anatomical underpinnings of respiratory modulations in brain dynamics. (p. 16)

Moreover, the reviewer's suggestion regarding nasal vs oral breathing is reflected in the 'outlook' section of the Discussion (see above).

2.2 The statement that the differences in modulation phase of aperiodic/periodic activity 'show at least partly differential dynamics' (discussion) or that these differences 'point towards a functional distinction...' are rather strong. Oscillatory activity is likely based on specific mechanisms which are distinct from those underlying non-oscillatory activity. If a common factor, let us say a modulatory afferent signal from primary olfactory regions, affects both patterns of activity, differences in the distribution of power versus phase may just reflect the different nature of the signal that has been analyzed. This would be a 'trivial' (though potentially important) difference, but from the point of respiration rooted in one identical mechanism. This discussion needs no be reflected extensively in the paper, but in my view statements about differences in mechanisms or biological/cognitive consequences should be toned down.

We fully agree with the reviewer's argument that the mechanisms underlying oscillatory coupling are different from non-oscillatory activity, which is precisely the point we wanted to make. Stating 'at least partly differential dynamics' is not an interpretation at all, but just meant as a summary of the findings shown in Fig. 4. However, in order to avoid an overly strong interpretation in terms of 'functional distinctions', we have rephrased the corresponding sections in the Abstract and Discussion (in addition to the changes made in response to the reviewer's previous point). They now read as follows:

Moreover, differential temporal dynamics in their coupling to non-oscillatory and oscillatory activity raise the possibility of a functional distinction in the way each component is related to respiration. (p. 2)

Moreover, differential temporal dynamics in their coupling to non-oscillatory and oscillatory activity raise the question whether there is a functional distinction in the way each component is related to respiration. (p. 13)

3. Presentation: While the methodology and logical flow of arguments are very clearly presented, few original data are shown and some parts of the figures are difficult to resolve.

3.1 Original data: I am not sure whether a raw trace can illustrate phase coupling of 1/f noise or periodic oscillations. If this turns out impossible, at least 1/f plots with different slopes could

be shown, maybe with multiple curves color-coded according to phase. In any case, the present data are mostly highly processed, hampering intuitive understanding.

We appreciate the reviewer's suggestion to show more unprocessed data. Following this suggestion, we have added a new panel to Fig. 2 showing - for one randomly chosen participant from the MEG sample - $1/f$ slope estimates and respiratory phase over time:

Fig. R5. Exemplary overlay of ROI-average normalised $1/f$ slope estimates and respiratory phase over time for a randomly chosen participant from the MEG sample. Transparent black lines show single-sensor slope estimates, solid black line shows mean slope across ROI sensors. Blueish-green scatter plot shows respiratory phase. Shown as panel b in Fig. 2 of the revised manuscript.

3.2 Figures: First, several panels like Fig. 1b, Fig. 4b,c and others are very small (not a major issue in times of electronic publishing, but tedious for the reader). Second, the unit and scale of the radial axis of polar plots is often unclear to the (non-expert) reader, and the insets in Figure 2 are very small and not sufficiently described. Third, the size of the differences between modulation of aperiodic and periodic activity, presented in Fig 4, is difficult to judge for most readers. Any way to highlight prominent cases (or, as a contrast, cases with small differences) or to make the deviation between $1/f$ and oscillation power more plastic/visible to the reader would be very welcome. Again, the parameters for this difference, as depicted in Fig. 4b (0-3) and 4c (-0.2 – 0.2) are not clear for most readers.

We thank the reviewer for his helpful comments on our figures. During the revision of our figures in response to the reviewers' collective suggestions, we have included the reviewer's remarks to i) scale up several panels, ii) add information with regard to units and scales, and iii) amend illustrations of differences. In response to an editorial comment, we have replaced the original bar plot in Fig. 4 with a violin plot. Furthermore, the reviewer's remarks have made clear that MAD as a new metric was difficult to evaluate for the reader. Consequently, we adapted the figure panel to now show individual U^2 statistics (as reported for the group level in the main text) and have included examples for strong and weak modulations in the updated Fig. 4:

Fig. 4. Source-level ROI modulation of 1/f slope and accumulated oscillatory power in the MEG. **a**, For each cortical node of the RMBO network², we show the group-level average courses of 1/f slope (left polar plots) and accumulated spectral power after removal of the aperiodic component (1 - 40 Hz; right polar plots) over the respiratory cycle. For ROIs comprising more than one parcel from the HCP atlas, slope and power courses are superimposed for each individual parcel. In case the respective LMEM did not indicate significant association with either slope or power within any given parcel, the corresponding time series are drawn with reduced opacity. For significantly associated parcels only, red and blue triangles indicate respective maxima and minima over the respiratory cycle (see legend). **b**, Polar histogram shows phase differences between the circular means of slope and power courses across parcels. The clear majority of non-zero phase differences were corroborated by Watson's U^2 statistics indicating distinct temporal dynamics of slope and power modulations within each parcel (see main text). **c**, Centre: For each ROI, we show violin plots of single-participant U^2 statistics (slope/power vs uniform distribution, respectively). Higher U^2 statistics indicate stronger non-uniformity. Left: Exemplary slope modulations of SMA and OFC (normalised across respiration phase for illustration purposes). SMA was more strongly non-uniform overall (see centre), which is visible from both stronger maxima (Δ_{\max}) and minima (Δ_{\min}) compared to OFC. Right: Exemplary power modulations of OFC and I. INS. For both ROIs, we superimpose power spectra (with 1/f removed) from all 60 phase bins. Overall, power varied more strongly with respiratory phase within OFC compared to I.INS (see centre). TPJ = temporo-parietal junction, ITG/STG = inferior/superior temporal gyrus, ANG = angular gyrus, FEF = frontal eye-field, TP = temporal pole, SMA = supplementary motor area, OFC = orbitofrontal cortex, CUN = cuneus, LS = lingual sulcus, ACC/PCC = anterior/posterior cingulate cortex, INS = insular cortex. In = inspiration, ex = expiration.

The Results and Methods sections have been revised accordingly:

For both slope and accumulated power, Fig. 4c shows ROI-wise distributions of subject-level U^2 statistics for each parcel as an estimate of respective non-uniformity (centre panel). Exemplary comparisons between ROIs are shown for slope ~ respiration (left panel) and power spectra ~ respiration (right panel). (p. 11)

Finally, we characterised the strength of slope and power modulations for each of the $n = 10$ RMBO ROIs by means of U^2 statistics. For each participant, we conducted Watson's U^2 tests of the (potentially multimodal) courses of $1/f$ slope and accumulated power within each ROI against a uniform distribution. Here, higher U^2 values indicate stronger non-uniformity within a given parcel, which allowed us to characterise RMBO ROIs by their respective group-level U^2 distributions (as shown in Fig. 4c). (p. 28)

Specific comments (not ordered by priority):

4. The two locations (Münster, Leipzig) are frequently described as 'sites'. This is confusing to the naïve reader. Within the context of EEG/MEG and with the focus on the parieto-occipital region, many readers will associate 'site' or 'recording site' with a specific spot on the subject's head. I suggest using another word, e.g. referring to the two distinct teams or groups.

We thank the reviewer for raising this point. In order to avoid any ambiguity, we now consistently refer to 'labs' instead of 'sites' throughout the manuscript.

5. The content of Table 1 can be incorporated into the legend of Fig. 1.

Following the reviewer's suggestion, Table 1 has been removed and its content is now shown in the legend of Fig. 2 (which is the corresponding figure).

6. Figure 1 and the Methods section indicate that power spectra are constructed from 1-40 Hz. The sliding window (time window 1 s, time steps 0.25 s) will probably not allow to measure reliably at low frequencies such as 1 Hz. Can you please comment and/or clarify?

We agree that the underlying FFT will probably not perform ideally for frequencies as low as 1 Hz (i.e., with a single oscillation per sliding window). However, we did not investigate power spectra below 2 Hz, at which point the Fourier transform becomes much more reliable (and even more so at higher frequencies). Kindly note that the delta band has been removed from the single-band power analysis shown in Fig. 5 (previously Suppl. Fig. S1).

7. The long discussion of alpha rhythms in the first paragraph of the introduction is distracting and may be partially moved to the discussion.

Thank you, the Introduction section has been revised accordingly and the alpha-related paragraph has been moved to the Discussion.

8. The discussion, on the other hand, could be shortened from ~4.3 to 2-3 pages at most.

We appreciate the reviewer's comment and have shortened the Discussion section to the best of our abilities without compromising central arguments.

9. 'IBB' and 'MPI' should not be used as figure labels (e.g., left and middle panels of Fig. 2). These are institutions, not biological entities!

Thank you, the respective figure labels have been revised.

10. Section 'Periodic and aperiodic components of...', second paragraph: were null beta weights really constructed with 500 tests or, rather, with 5000?

For the higher-dimensional null distributions (i.e., channel x time or parcel x time), surrogate LMEM weights were initially indeed computed 500 times, as 5000 random iterations would have resulted in ~15 consecutive days of computation time. With the revised methodology and adjusted codes, we now consistently use 5000 null iterations throughout all analyses.

11. In the same paragraph (see comment #10) the use of 'slope and power' for aperiodic and periodic activity measures, respectively, should be made more explicit to avoid confusion.

Thank you, the respective paragraph has been rephrased:

In keeping with our sensor-level analyses, we determined significance of respiration phase-locked changes in aperiodic slope and oscillatory power by means of LMEMs within each parcel of interest. (p. 11)

Reviewer #3

Summary:

In this paper, Kluger and colleagues study the relationship between respiratory cycles to fluctuations in cortical dynamics. Specifically, they assess whether the aperiodic exponent of the power spectrum of M/EEG recordings—as a surrogate of cortical excitation-inhibition balance—vary dynamically with the phase of respiratory cycles, using a novel time-frequency decomposition and parameterization tool (SPRINT). They find that, in both EEG and MEG data and across the cortex (and specifically in parietal-occipital regions), aperiodic exponent fluctuates with respiration peak and troughs. They repeat the analysis under several conditions, including per-electrode level, HCP parcellation-level, etc., and find that the effect is robust (along with appropriate statistical testing). Finally, they perform the same analysis but with aperiodic-removed oscillatory power, and show that aperiodic and oscillatory components have different respiration phase preference, suggesting different mechanisms of modulation.

Overall, I find the study to be well motivated given the gap in respiration-brain coupling literature, and also clearly written / presented. The application of SPRINT in this context is novel, and contributes interesting data on how respiration may be related to distinct neural processes in the aperiodic and oscillatory components of M/EEG. I have several concerns, however, regarding signal processing decisions, as well as interpretations and the lack of data on the modulation of aperiodic-removed alpha oscillation, especially in relation to the authors' previous work [ref 9]. Some other minor comments / suggestions are also included below.

- a note on nomenclature: I don't mean to be a prescriptivist, but I find the authors' usage of slope and exponent to be a bit confusing. In the specparam description of the M/EEG power spectrum, the aperiodic component follows a power law, i.e., power scales inversely with frequency, with a certain scaling exponent, $P \sim 1 / (f^\alpha)$. Alpha here would be the exponent of the aperiodic spectrum. This can be reformulated as a linear relationship in log-log, i.e. $\log P \sim -\alpha * f$, and alpha describes the slope of that line. Throughout the manuscript, the authors refer to this quantity, alpha, as the aperiodic or 1/f slope, and refer to changes as steeper or flatter slope, which is not problematic at all but perhaps warrants a short explanation upfront about the loglog transform. What is problematic is that the phrase "slope exponent" appears many times in the paper, which are basically two words that mean the same thing, but I take it to mean essentially "the magnitude of the aperiodic slope". I think everywhere it says "slope exponent" can just be replaced with "slope". For the rest of the review, I will follow the authors convention and use slope to mean the exponent of the power law aperiodic spectrum, where steeper slope refers to a larger exponent.

We thank the reviewer for his comprehensive argument. Following the reviewer's suggestion, we have replaced all instances of 'slope exponent' with 'slope' for the sake of accuracy.

- another note, please include page number and line number, it's very difficult to refer to specific places in the manuscript otherwise (unless this is introduced in the submission process, if so please complain), and it will be equally difficult for the authors to find where I'm talking about in the manuscript.

Thank you, the revised manuscript now includes page and line numbers.

1. clarifications and concerns regarding signal processing:

- if I understand correctly, the main pipeline is as follows: SPRINT outputs aperiodic and periodic parameter estimates from short time Fourier transforms with 250ms step length, resulting in a 4Hz slope time series (and oscillatory power estimates). The respiratory signal is recorded and peaks/troughs extracted, and the phase is linearly interpolated between peak to trough and trough to peak (I assume continuously?). Then, this respiration phase signal is "sampled" wherever there is a slope value, where respiration phase is first split into 60 bins, and the "effective respiratory phase" is a time series that is sampled at identical time points as the slope time series (so 4Hz), with value taken as one of the 60 possible bin values. Subsequent analyses average slope values that belong to the same phase bin, where permutation tests are also conducted. If this is correct, I have several concerns and questions:

- where it says "we collected all SPRINT outputs computed at respiration angle of $\omega \pm \pi/10$ ", I'm not really sure what this means? Does it mean the bin center is at ω , but the bin is actually $\pi/5$ wide, such that the 60 phase bins are overlapping?

The reviewer's description of our pipeline is completely accurate and our phase bin computation indeed uses overlapping windows. In order to avoid ambiguity in this respect, the manuscript has been revised throughout where appropriate.

- Similarly, "we computed individual bin-wise averages....,yielding quasi-continuous 'phase courses'...", and later, "we finally binned all SPRINT outputs into $n=60$... and computed the bin-wise average slope fit to yield a null time series...". Bin-wise averaging means to average

all slope values that fall within that bin, i.e., all slope values that share a corresponding phase bin “ID”? If so, doesn't this mean it's collapsed across time already, so how is a “phase course” and “null time series” arrived at? My naive expectation would be that there is simply two time series, both at 4Hz and at matching time points, where the slope is just the SPRINT output for that STFT window, and respiratory phase is the linear interpolation queried at that time point? Please clarify on these two points above.

We apologise for the apparent confusion caused by our terminology. The reviewer's assumption with regard to our methodology is entirely correct and it has become clear that our phrasing was not ideal. To clarify, by ‘phase course’ we simply meant slope ~ phase (i.e., not a time series with slope values, but a series of phase bins with slope values). The inept term ‘phase course’ has been removed from the manuscript and the corresponding paragraph now reads as follows:

At this point, we computed individual bin-wise averages of 1/f slope and periodic spectra, yielding mean slopes and spectra for each participant quasi-continuously over respiratory phase. (p. 24)

Similarly, we acknowledge that the term ‘null time series’ was an imprecise description of our approach. The term was merely to state that we computed slope ~ phase for surrogate data on the phase dimension. For the sake of clarity, we have rephrased the paragraph as follows:

In keeping with the approach above, we finally binned all SPRiNT outputs into $n = 60$ equidistant, overlapping phase bins covering one entire respiratory cycle ($-\pi$ to π) and computed the bin-wise average slope fit to yield a ‘null time series’ of 1/f slope over surrogate respiration phase. For each participant, this procedure was repeated 5000 times and resulted in a null distribution of 5000 surrogate 1/f slope estimates x 60 phase bins. (p. 26)

- related, I really like Fig 1 that shows a schematic of the analysis pipeline. I think it could be a little clear. For one, the respiration signal has an arrow into the SPRINT box, but it doesn't really influence the M/EEG analysis in anyway if I understand the algorithm correctly. If anything, where there is slope estimates determines where the respiratory phase is sampled, such that the two form two parallel time series sample at the same time points. I believe Fig 1b shows this? Maybe one way to make it clearer is to show draw an arrow from the top part of a (the resp signal) into the “extract respiratory phase” part in b, and a parallel arrow from SPRINT to the top part of b (with some rearrangement). This is just a suggestion on how to potentially make it clearer to avoid confusions such as my two points above.

We truly appreciate the reviewer's suggestions with regard to the optimisation of Fig. 1. The respiratory signal indeed does not inform the SPRiNT algorithm, so we have removed the arrow as suggested. In order to make the parallel streams of M/EEG and respiratory data more prominent, we have reordered the panels to now depict the SPRiNT section on the very top of the figure (and away from the respiratory trace). As suggested by the reviewer, arrows now link the SPRiNT panel to the extraction of time-resolved slopes/spectra (top of panel b) and the respiratory time course to the phase binning panel. Finally, the middle panel of Fig 1b has been rephrased to ‘match with respiratory phase’, as the phase extraction itself seemed

redundant in the revised order.

Further changes have been made to both panels in response to comments by Reviewer 1. The 'output' section is now similar to the original publication by Wilson et al. (eLife, 2022) and the illustrations of the time-resolved slope fits and spectra have been changed.

We hope that Fig 1 has gained both clarity and accessibility and we welcome any further suggestion the reviewer might have.

Fig. 1. Synopsis of acquired data and applied methods. **a**, In two labs, we simultaneously acquired nasal respiration as well as eyes-open, resting-state MEG (IBB, Münster) and EEG data (MPI, Leipzig) in continuous 5-min recordings. After preprocessing, single-sensor/-channel M/EEG data (middle panel) were subjected to the *SPRiNT* algorithm²⁶ (top). Here, using a moving-window approach (window length = 1 s, 75% overlap between neighbouring windows), spectral components of neural time series are estimated using a Fast Fourier transform. These frequency-domain data are then parameterized using the *specparam* algorithm²⁵ which yields both aperiodic and periodic components of neural activity in that time window. Repeating this procedure along the entire recording thus yields time-resolved fits of the aperiodic 1/f slope as well as time-resolved periodic spectra ranging from 1 - 40 Hz. Respiratory phase was computed using two-way interpolation (int) of the normalised raw respiration signal (peak-to-trough, trough-to-peak; bottom). **b**, For each time point used as a moving-window centre in the *SPRiNT* algorithm, we then extracted the corresponding respiratory phase. This allowed us to sort all time-resolved slope fits and periodic spectra according to the respiratory phase at which they were computed. In keeping with previous work^{2,9} we finally partitioned the respiration cycle into $n = 60$ equidistant, overlapping phase bins and computed bin-wise averages of slope fits and periodic spectra. This approach thus yielded quasi-continuous, respiration phase-resolved courses of both periodic and aperiodic components of brain activity for each sensor/channel within each participant.

- I'm not familiar with the respiration DSP literature, but is it standard practice to find the peak and trough and simply linearly interpolate between the two for phase estimates? Why not just use the analytic phase (via Hilbert transform) since it's a relatively narrowband signal? The latter is what one would usually define as "phase", and compared to that, the former method could introduce substantial errors in phase estimate, and therefore binning, which could significantly affect the result of the study. Specifically, for a non-sinusoidal signal, the distribution of phases is not uniform around the circle if divided into equal bins. Alternatively, one could bin such that each of the 60 bins have the same number of samples, and would be equally justifiable. In fact, the authors should show the distribution of phases, i.e. a histogram with the 60 defined bins, and, importantly, in general discuss why linearly interpolating phase would be preferable to using an actual estimate from Hilbert transform. I would also

recommend repeat at least a subset of the analyses using the Hilbert phase to assess whether such a choice would impact the result significantly.

The reviewer's comment addresses several critical aspects with regard to the preprocessing of respiratory data and we appreciate the opportunity to discuss them here briefly. Choosing the proper method for respiratory phase extraction has been a methodological focus of our work for the past years. Several algorithms such as the double interpolation implemented here (Pikovsky et al., 2002) or the so-called *protophase* projection (Kralemann et al., Phys Rev E 2008; Kralemann et al., Nat Comms 2013) have been developed specifically to process non-sinusoidal signals. Across multiple studies, labs, and acquisition methods for respiratory data, we have consistently observed that Matlab's Hilbert transform is more prone to yield artificial phases the more non-sinusoidal participants' respiratory patterns are, whereas the double interpolation method is more robust. In collaboration with Micah Allen's lab at Aarhus University, we are currently preparing two manuscripts demonstrating the disadvantages of using the Hilbert transform on respiratory data.

To briefly illustrate the accuracy of both Hilbert transform and the interpolation approach, the following figures show excerpts of respiratory data for two randomly selected participants. Notice how the Hilbert transform misplaces the onsets of respiratory cycles (i.e., $\pm \pi$):

Fig. R6. Exemplary excerpt of respiratory data and phase extraction for a randomly selected participant. Red dashed line shows the normalised respiration trace. Purple line shows respiratory phase extracted with the Hilbert transform, green line shows phase extracted by the double interpolation approach used in the manuscript. Yellow markings show phase offset between Hilbert and double interpolation.

In addition to imprecise detection of cycle onsets, the Hilbert transform tends to falsely detect respiratory 'peaks' and wrongly mark cycle onsets accordingly. The following data are taken from a different participant than shown above:

Fig. R7. Exemplary excerpt of respiratory data and phase extraction for a second randomly selected participant. Red dashed line shows the normalised respiration trace. Purple line shows respiratory phase extracted with the Hilbert transform, green line shows phase extracted by the double interpolation approach used in the manuscript. Yellow box shows rapid succession of respiratory cycles artificially marked by the Hilbert transform, yellow markings show phase offset between Hilbert and double interpolation.

In light of these volatilities when applied on respiratory data, we hope to have convinced the reviewer that re-running analyses based on Hilbert would not be meaningful.

The question regarding the binning procedure is equally important and eventually comes down to a catch 22 where one can either have the same number of observations within each bin (but different temporal resolutions across trials) or the other way around. Given that i) we have sufficient statistical power within each bin and ii) our research question is focussed on temporal (i.e., phase) characteristics, we have consistently chosen to keep the duration of the phase bins fixed and allow for variation in sample numbers across bins. Following the reviewer's suggestion, we have added new Supplementary Figures showing bin-wise numbers of observations (i.e., SPRiNT outputs) per participant in both MEG (Fig. S3) and EEG data (Fig. S4):

Fig. S3. Individual numbers of observations per respiration phase bin in the MEG sample.

Fig. S4. Individual numbers of observations per respiration phase bin in the EEG sample.

- in general, I'm a bit confused at how the shuffling was done for the permutation tests. When it says "we shuffled respiration time course", does it mean shuffling the raw signal? the interpolated phase value? or something else? My naive guess is that the 4Hz signal (either the resp phase or slope) is shuffled across its time index, in which case this might be described more clearly? Furthermore, since naively shuffling destroys temporal correlation, the authors might consider using a method that creates a null time series while preserving autocorrelation (e.g., circular shifting, IAAFT, etc.)

We thank the reviewer for bringing our attention to what was obviously an insufficiently clear account of our null distribution computations. The reviewer is correct that our previous randomisation procedure involved shuffling of the extracted respiratory phase along the time index. This was done to construct null distributions of $1/f$ slope at two instances during our analyses: In our first analysis, we used these null distributions to evaluate *at which phases* ROI-average slope was significantly modulated (concentric scatter plots of black dots in Fig. 2). In the subsequent whole-scalp analyses, we determined the effect size of empirical slope modulations relative to these null distributions (topography of z-values in Fig. 3c). All remaining analyses - most importantly, the LMEMs determining overall significance of ROI-average (Fig. 2), whole-head (Fig. 3), or source-space modulations (Fig. 4) - did not involve shuffling of the raw respiratory signal (see Methods for details).

Nevertheless, as pointed out by both Reviewers 1 and 3, our approach for computing 'null slopes' did not preserve the temporal autocorrelation of the original signal and conceivably affected the results shown in Fig. 2 and 3. Hence, in order to construct a more adequate surrogate distribution, we followed the reviewer's advice and used IAAFT (instead of shuffling) of the original signal. While there are slight changes in the timing and absolute size of the modulation effects compared to the initial submission (see Fig. 2), the overall pattern of results remains unchanged. Figs 2 and 3 as well as the corresponding paragraphs have been revised accordingly. Moreover, we carefully revised the description of our approach in the Methods section, which now reads as follows:

To test whether $1/f$ was significantly modulated over the ROI sensors, we computed $k = 5000$ random iterations of the LMEM by shuffling bin-wise $1/f$ slope fits on the individual level. This way, any meaningful relations between $1/f$ slope and respiratory phase were removed, thereby specifically testing the hypothesis that $1/f$ slope changes significantly with respiratory phase. For each of these iterations, we once again computed the phase vector norm from the resulting beta weights, yielding a random distribution of $k = 5000$ 'null vector norms'. Significance of the empirical vector norm was determined by its percentile relative to this null distribution.

As shown in Figs. 2 and 3, for the sensor-level analyses, SPRiNT parameterisation was conducted on individual MEG sensors (Münster) and EEG channels (Leipzig). For the analysis of $1/f$ slope modulations within the parieto-occipital ROI, we computed mean SPRiNT estimates across the corresponding sensors or channels, respectively. LMEMs were then computed on ROI-average estimates for $1/f$ slope ~ respiration phase for each lab individually (Münster, Leipzig) as well as for the pooled data from both labs (see Fig. 2). For the whole-scalp analyses shown in Fig. 3, LMEMs were computed on single-sensor/channel SPRiNT estimates.

Having established an overall significant influence of respiratory phase on $1/f$ slope with the LMEM approach described above, we next aimed to characterise *when* (i.e., at which respiratory phases) $1/f$ slope was significantly steeper or flatter. To this end, we implemented a permutation approach as follows: For each participant, we constructed a surrogate respiration time series using the iterated amplitude-adjusted Fourier transform (IAAFT; Theiler et al., 1992). In contrast to shuffling the respiration time series, this iterative procedure preserves the temporal autocorrelation of the signal, which is critical for constructing a fitting null distribution for permutation testing. From these IAAFT-transformed respiration time series, we extracted the surrogate respiratory phase values corresponding to each slope estimation from SPRiNT. In keeping with the approach above, we finally binned all SPRiNT outputs into $n = 60$ equidistant, overlapping phase bins covering one entire respiratory cycle ($-\pi$ to π) and computed the bin-wise average slope fit to yield a 'null time series' of $1/f$ slope over surrogate respiration phase. For each participant, this procedure was repeated 5000 times and resulted in a null distribution of 5000 surrogate $1/f$ slope estimates x 60 phase bins. (pp. 25-26)

- there were a few analyses that involved averaging across channels/sensors/sources before submitting the signal to SPRINT. In general, and especially when oscillatory signals are concerned, I would heavily recommend against averaging the raw signal across space, as it may result in destructive interference of a shared oscillation, resulting in a artificially lower estimate of power, especially if the oscillation itself is a traveling wave (as has been shown in many papers now). Similarly, it's not known how averaging would affect the broadband spectrum, but in general one can expect it to affect one part of the spectrum preferentially, e.g., averaging over white noise channels reduces overall power/variance, which could lead to decreasing power at higher frequencies (near the noise floor), hence affecting the slope estimate out of any specparam methods. Since in the per-electrode analysis, all the relevant slope and oscillation estimates were provided from SPRINT anyway, I would definitely recommend averaging at the slope estimate level, which could even result in less variance in the ROI slope estimate.

We thank the reviewer for this insightful comment. When directly comparing the MEG group-level mean slope ~ respiration over parieto-occipital MEG sensors (as a test case), it became

apparent that the averaging did indeed influence our slope estimates (albeit only slightly). As a consequence, we have recomputed the analyses shown in Fig. 2. Here, we now use the single-sensor/channel output and compute the ROI average on the slope estimate level (i.e., after SPRiNT was applied to single-sensor/channel spectra). As before, both individual data sets as well as the pooled data showed significant modulation of 1/f slope over the respiratory cycle. The precise phases of significant slope increases and decreases slightly changed, as shown in the revised Fig. 2:

Fig. 2. Group-level respiration-locked modulation of aperiodic 1/f slope over parieto-occipital sensors. **a**, Respiration phase was extracted for a total of $n = 60$ overlapping phase bins covering the entire respiration cycle between subsequent inhalations. Overall significance of respiration phase-locked modulations was assessed by means of permutation statistics: For each data set, the combined LMEM beta weight for the phase vector norm (i.e., $\sqrt{\sin^2 + \cos^2}$) is shown in light blue against a distribution of null betas ($k = 5000$, top right corners). Within each polar plot, coloured dots show respiration phase-dependent 1/f slope gained from SPRiNT computations on periodic spectra over parieto-occipital MEG sensors or EEG channels, respectively. Concentric scatter plots (black dots) indicate null distributions of group-level mean exponent values computed for each phase bin ($k = 5000$ iterations). Solid black lines indicate the 5th and 95th percentile of each bin's null distribution. In the MEG sample, 1/f slope was flattened during the inspiratory phase (around -70°) as well as around the inspiration-to expiration transition (-9° and 15° to 21° ; blue markings). 1/f slope was steeper during the expiration phase (82° to 88°) and around the expiration-to-inspiration transition (-174° to 162° ; red markings). Similarly, the EEG sample showed flatter slope after the inspiration-to-expiration transition (9° to 46° and around 58°) and steeper slope around the expiration-to-inspiration transition (-162° to 131°). In = inspiration, ex = expiration. **b**, Exemplary overlay of ROI-average 1/f slope estimates and respiratory phase over time for a randomly chosen participant from the MEG sample. Transparent black lines show single-sensor slope estimates, solid black line shows mean slope across ROI sensors. Blueish-green scatter plot shows respiratory phase. **c**, Correlation between mean 1/f slope and breathing rate (top) as well as breathing depth (bottom) for MEG (left) and EEG data (right). Neither respiratory parameter was significantly correlated with overall steepness of the aperiodic component during rest. **d**, For the analysis of normal vs deep breathing, we revisited a previously published MEG data set ³. For both breathing conditions, we show the respiratory trace of a single participant (left) and median group-level breathing depths (right). As instructed, participants were breathing deeper during the deep (vs normal) breathing condition ($z(27) = 6.19$, $p < .001$). **e**, Using the same preprocessing and ROI analysis pipelines as in the previous analyses, we tested whether the strength of 1/f slope modulations over posterior sensors differed between normal (i.e., automatic) and voluntary deep breathing. Indeed, the individual ranges of slope ~ respiration phase were consistently greater for deep (compared to normal) breathing ($z(27) = 3.02$, $p = .003$). **f**, Comparison of parieto-occipital 1/f slope ~ respiration phase during nasal (left) and oral breathing (right), collected in a follow-up control study (see Methods for details). Using the identical experimental setup and analyses as in the original MEG data, LMEMs confirmed a significant influence of respiratory phase on 1/f slope for both breathing conditions (see histogram insets). 1/f slope ~ respiration courses were similar to our previous MEG and EEG results. **g**, Individual ranges of slope ~ respiration (as a measure of modulation strength) did not differ between conditions ($z(25) = 0.80$, $p = .426$).

The Methods section has been revised accordingly and now reads as follows:

As shown in Figs. 2 and 3, for the sensor-level analyses, SPRiNT parameterisation was conducted on individual MEG sensors (Münster) and EEG channels (Leipzig). For the analysis of 1/f slope modulations within the parieto-occipital ROI, we computed mean SPRiNT estimates across the corresponding sensors or channels, respectively. LMEMs were then computed on ROI-average estimates for 1/f slope ~ respiration phase for each lab individually (Münster, Leipzig) as well as for the pooled data from both labs (see Fig. 2). (p. 25)

- it would be great to see some examples of the power spectrum in loglog, to see if a linear fit over the whole frequency range is appropriate. In many cases, we're seeing that the exponent estimate is biased by the shifting plateau (or knee) of the spectrum, even without real spectrum changes (e.g., Gao et al 2020 elife). I know this sounds like I'm passive aggressively suggesting for the authors to cite our new papers that design evermore spectrum features (it's really not), but in all honesty this is just a suggestion to ensure that the correct interpretation is made wrt E-I balance, and it would also be interesting if timescale is also affected by breathing. On that note, the authors should cite Trakoshis et al 2020, elife, which has a much better computational model linking E-I changes to spectral slope.

We welcome the reviewer's suggestion to provide a more detailed account of the power spectra. We have now included log-log depictions of each participant's spectra in new Supplementary Figs. S1 (MEG) and S2 (EEG):

Fig. S5. Individual MEG power spectra (averaged across sensors within the parieto-occipital ROI) in log-log transform. For each participant, we show power spectra in the frequency range from 1 Hz to 40 Hz for each of the 60 respiratory phase bins (colour-coded).

Fig. S6. Individual EEG power spectra (averaged across channels within the parieto-occipital ROI) in log-log transform. For each participant, we show power spectra for each of the 60 respiratory phase bins (colour-coded).

This comprehensive overview shows that - with few exceptions - the linear fit is relatively well-suited to estimate 1/f slope in our data sets. While, as is to be expected, there are clear peaks visible in the power spectra (most prominently in the alpha band), there are no clear knee

points visible. This is most likely due to the fact that we are using a frequency range of 1-40 Hz for fitting; the knee frequency could conceivably be just above this range.

Finally, we thank the reviewer for suggesting the Trakoshis paper (eLife, 2020), which is now referenced in the manuscript.

2. clarifications on results:

- given that the authors' previous publication investigated alpha power and respiratory phase (presumably without specparam or SPRINT), I am very curious whether aperiodic-removed alpha power modulation is still associated with specific phases of respiration. Related, I don't really understand the justification to sum over the entire periodic spectrum, and while the authors did the analysis on per-band power, nothing is mentioned on that front other than that it's in Supp Fig 1. How come?

We appreciate the reviewer's remarks with regard to single-band power modulations, which was indeed the focus of previous work from our lab. The study in question (Kluger et al., eLife 2021) had used a near-threshold visual perception task with well-known implications on alpha power, which we then linked to respiratory phase (indeed without using parameterisation). In contrast, the present study (as the first foray into aperiodic dynamics and peripheral rhythms) exclusively used resting-state recordings, which complicates intuitive across-study comparisons. Nevertheless, the reviewer's comments have made it clear that our presentation would profit from a more comprehensive description of our oscillation results, particularly with regard to single-band power.

First, as for the reviewer's question regarding aperiodic-removed alpha power, we did indeed find significant modulation of 'fooofed' power spectra in the alpha band. In fact, alpha was consistently modulated across ROI parcels, as can be seen below:

Fig. 5. Source-level ROI modulation of band-specific oscillatory power in the MEG. **a**, For each cortical node of the RMBO network, we show the group-level average courses of band-specific spectral power after removal of the aperiodic component over the respiratory cycle. For ROIs comprising more than one parcel from the HCP atlas, power courses are shown for each individual parcel. In case the band-specific LMEM did not indicate significant modulation of the periodic component within any given parcel, the corresponding power series are not shown. **b**, Exemplary ROI-average spectra of anterior cingulate (top) and left insula (bottom), colour-coded according to

respiratory phase. c, Group-level median U^2 statistics of each parcel testing single-band power ~ respiration phase against a uniform distribution. Higher values indicate stronger deviation from uniform distribution.

As outlined above, we were cautious not to draw overly strong conclusions regarding 1/f-removed alpha power due to the contextual differences between the present resting state and previous task-related measurements. Thus, our band-specific analysis of 'fooofed' power was to be a descriptive first step to prompt dedicated future work. The measure of cumulated power (1-40 Hz) was chosen as a criterion for the residual oscillatory power remaining after the removal of the aperiodic component. Following the reviewer's comments, however, we were eager to illustrate the effects of respiration on single-band power more comprehensively. To this end, we have revised the previous Supplementary Fig. 1 and included it in the main text as a new Fig. 5. In keeping with the presentation of 1/f slope results in Fig. 4, this new figure now shows band-specific power for theta, alpha, and beta frequencies over the respiration cycle. For easier interpretation, we have removed delta (which was limited by SPRiNT's frequency limits, see a related comment by Reviewer 2) and the somewhat unspecific 'beta 2' band (30 - 40 Hz). We have added exemplary ROI-average power spectra plotted across phase bins to show differential modulations particularly within alpha and beta (see Fig. 5b). Finally, we adopted the comparison based on U^2 statistics taken from the slope results to characterise the strength of modulatory effects across parcels and frequency bands (see Fig. 5c).

Finally, single-band modulations of 1/f-removed theta, alpha, and beta power within the RMBO network are shown in Fig. 5. Out of 23 parcels, band-specific LMEMs revealed that theta power was significantly modulated in 17 parcels, alpha power in 22 parcels, and beta power in 21 parcels (see Fig. 5a and Methods for details). While band-specific oscillatory modulations were not the main focus of this paper, we provide this characterisation to motivate hypothesis generation for future work. (p. 13)

- in Figure 2, it looks like the significant regions of the radial histogram is slightly rotated between EEG and MEG, especially for the slope decrease regions. In addition, it looks like for the significant increase and decrease are (roughly) pi degrees apart in the MEG dataset, but not in the EEG. Are there explanations or discussion for this?

The slight rotation between MEG and EEG ROI results has further decreased after introducing the new averaging approach the reviewer previously suggested (see revised Fig. 2 above). At present, the most prominent difference between both data sets is the significant decrease visible in the MEG data around the midway point of inspiration, which is not significant in the EEG data. As there is both a clear (albeit non-significant) trend in the EEG data as well as a significant (and even more substantial) slope decrease when both data sets are pooled, this difference most likely is not meaningful, but rather a mere difference in power/SNR between both methods. This is further corroborated by the fact that MEG and EEG slope ~ respiration courses did not differ significantly from one another (non-parametric two-sample test for circular data: $P(77) = 0.21$, $p = .651$).

- Comparing Figure 3a to figure 2, it looks like the histogram of mean phases (Fig3a) are rotated vs. the maximum bin-average slope magnitude (Fig2 red): is this meaningful, or due to the two different quantities computed? In general, do the circular distributions across these analyses agree with each other (statistically not significantly different)?

We agree with the reviewer that the perceived rotation between the histograms of Fig. 3 and mean slopes in Fig. 2 are most likely due to the different criteria (particularly the circular mean). As outlined in the manuscript, the circular mean might vary in accuracy depending on how far the underlying data differ from a unimodal distribution. Thus, distribution comparisons across the analyses should also be interpreted with caution (and were therefore not included in the manuscript). In order to address the reviewer's question, we conducted two control analyses: First, for the MEG sample, we compared the group-level mean slope course within the posterior ROI (Fig. 2) to single-sensor slope courses (Fig. 3). Specifically, for each sensor, we used a non-parametric median test (see main text) to assess whether this sensor's circular mean (across participants) differed from the ROI average (again, across participants). Out of 275 sensors in total, a meaningful difference in circular means was found for only 10 sensors. This consistency across analyses was supported by a second analysis in which we computed the optimal phase lag to maximise the correlation between ROI-average and single-sensor slope ~ respiration. For each sensor, we used the circshift function in Matlab to shift this sensor's slope course against the ROI average (shift range: $\pm \pi/2$) and compute the correlation between the two slope courses. Single-sensor optimal phase lags were determined as the shifts at which the correlation was maximised. The distribution of optimal phase lags is shown as a polar histogram in Fig. R8 below:

Fig. R8. Consistency analysis for slope courses between posterior ROI and whole scalp. Histogram shows the distribution of optimal phase lags maximising the correlation between ROI-avg and single-sensor courses of slope ~ respiration.

Testing both the circular mean ($p > .99$) and median against zero ($p = .358$) confirmed that the correlation between ROI-average and single-sensor courses of slope ~ respiration was maximised at zero lag, which again underscored the consistency across these two analyses.

- Figure 3b shows many spots, especially in the EEG dataset, where the mean phase is near 0 instead of π . Is there an explanation for this?

We assume that the different mean phases within each topography are at least in part caused by the circular mean measure itself. As we state later in the manuscript (kindly see pp. 9-10), the circular mean is not always optimally suited to describe multimodal distributions. As slope ~ respiration courses vary between different sensors/channels, the circular mean is conceivably more adequate for some sensors/channels than for others (e.g., those with clearly multimodal slope courses).

- Figure 4b shows a distribution of differences of mean phase, but is not weighted by the actual means nor the significance of the means, correct? In other words, many channels could have very small mean phases (i.e. near uniform circular distributions) for both slope and power, but

nevertheless result in a difference, is this be meaningful? What would happen if one only took histogram over parcels that were significantly non-uniform for either or both slope and oscillation? Apologies if I had missed where this was done.

We apologise for the lack of clarity regarding the histogram of differences in Fig 4b. As all of the $N = 23$ ROI parcels showed either a significant modulation of slope ($n = 18$) or power ($n = 19$), the histogram actually showed differences for significantly non-uniform parcels only. To avoid future misunderstandings, the manuscript has been revised as follows:

Of the 23 parcels of interest, 18 parcels showed a significant association between respiration and $1/f$ slope. Significant phase-locked modulations of oscillatory power (within frequencies from 1 - 40 Hz) were found for 19 parcels (see Fig. 4a). Overall, all 23 parcels showed either significant non-uniformity of slope or accumulated power. (p. 11)

- throughout the manuscript, it seems that the authors imply that breathing causally modulates slope (hence E-I balance), whereas the data only shows a correlation. This could be a misinterpretation on my part, but given the discussion on predictive processing and the volitional control of breathing, it seems that the authors have an implicit stance that breathing phasically modulates cortical dynamics. I don't believe the current data supports this, as opposed to, say, a task where participants are asked to breathe at controlled time points. I believe this warrants some qualification in the discussion section.

We thank the reviewer for suggesting a more cautious discussion of our findings in order to adequately reflect the associative (rather than causal) nature of the study. We acknowledge that our previous phrasing was overly liberal and have rephrased all sections of the manuscript accordingly.

For both MEG and EEG as well as for the pooled data, the empirical group-level beta weight for respiratory phase was significantly higher than the respective null beta weights (all $p < .001$), strongly indicating a significant overall association between respiratory phase and aperiodic neural signals over posterior cortices (Fig. 2). (p. 5)

Of the 23 parcels of interest, 18 parcels showed a significant association between respiration and $1/f$ slope. (p. 11)

In case the respective LMEM did not indicate significant association with either slope or power within any given parcel, the corresponding time series are drawn with reduced opacity. For significantly associated parcels only, red and blue triangles indicate respective maxima and minima over the respiratory cycle. (p. 12)

In the present data, the preferred phase differences between source-level periodic and aperiodic modulations within the RMBO network show at least partly differential dynamics between both components, suggesting a more complex coupling of neural signalling to the respiratory rhythm than previously assumed. (p. 15)

If $1/f$ slope and oscillatory power were similarly coupled to the respiratory rhythm, their respective courses over respiratory phase would not differ and the phase difference between their circular means would be (close to) zero. In contrast, systematically different distributions of slope and power would suggest differential dynamics in their respective coupling to respiratory phase. (p. 27)

As stated by the reviewer, ‘timed breathing’ tasks would indeed be required for a causal inference of respiration-locked aperiodic dynamics. To inspire future work, we have added this suggestion to the ‘Future Directions’ paragraph of the Discussion section alongside potential clinical applications and respiratory manipulations (e.g., nasal vs oral breathing):

The present findings open multiple avenues for future research, of which we want to briefly sketch three main directions. First, mechanistic advances could be made in a replicative study which includes a contrast of nasal vs oral respiration. The absence of phase-amplitude coupling driven by nasal airstreams during oral breathing could conceivably reveal to what extent aperiodic changes rely on and/or interact with oscillatory changes. While we speculate that respective changes in $1/f$ slope and oscillatory power could point to distinct modulatory pathways, targeted follow-up studies with sufficient statistical power are of critical importance to extend our initial results. Second, as outlined above, it would further be highly instructive to complement existing evidence of task-related $1/f$ slope changes^{23,54} by investigating their link to respiration in behavioural cognitive or perceptual paradigms. Timed breathing (in healthy participants or ventilated patients) or contexts involving different brain states, e.g. resting-state vs task, during attentional or arousal manipulation (including hyperventilation), or during sleep stages, could potentially unravel the link between breathing and fluctuations of (non-)oscillatory brain activity. (pp. 16-17)

REVIEWERS' COMMENTS

Reviewer #1 (Remarks to the Author):

I am appreciative of the efforts and consideration brought by the authors in response to my suggestions and comments. The points that required clarification are not better explained, including in the revised figures. I remain skeptical of the significance of the results reported in Figures 3 and 4.

I will defer to the other reviewers and the editors concerning the final recommendation for publication in the journal. My impression is that this is current and honest work that addresses significant questions in the field, but falls a bit short to bringing the data reported to a full home-run.

Reviewer #2 (Remarks to the Author):

Kluger and coworkers have made substantial changes and amendments to the manuscript. In particular, they included new data which, in my eyes, make an important contribution to the content. They also improved figures and multiple text passages.

I have no further concerns and think that this paper makes an important contribution to the methodology and understanding of brain-body interactions.

Reviewer #3 (Remarks to the Author):

I commend the authors for constructively and comprehensively engaging with my previous comments; the revision has satisfactorily addressed my concerns and suggestions. I especially appreciated the comparison to Hilbert-based phase extraction as I learned something new. All good from my side.
Richard Gao